# Regulation of DMSP organosulfur cycling in ubiquitous Roseobacter marine bacteria

Hui-Hui Fu [ID][1,7], Ming-Chen Wang [ID][1,7], Zhi-Qing Wang[1], Yu-Han Sang[1], Zhen-Kun Li[1], Fei-Fei Li[1], Jia-Rong Liu[1], Qi-Long Qin[2,3], Xiao-Yu Zhu[1,4], Na Wang[1], Jin-Jian Wan[1], Zhao-Jie Teng[2,3], Wei-Peng Zhang[1], Andrew J Gates [ID][4,5], Chun-Yang Li [ID][1,3], Jonathan D Todd [ID][1,4,5,6 ✉] & Yu-Zhong Zhang [ID][1,2,3 ✉]

## Abstract

Dimethylsulfoniopropionate (DMSP) catabolism by marine Roseobacters is important for global biogeochemical cycling and the climate. Many Roseobacters contain competing DMSP demethylation and cleavage pathways, but only cleavage produces the climate-cooling gas dimethylsulfide. Here, we identify the "switch" regulator in Roseobacters, DmdR, which transcriptionally represses demethylation (*dmdA*, encoding DMSP demethylase), cleavage (*acuI*, encoding acryloyl-CoA reductase) and oxidative stress protection (*dmdEF, dinB*) genes under low intracellular DMSP levels. Increased DMSP levels lead to DMSP cleavage and accumulation of cytotoxic cleavage product acryloyl-CoA. Acryloyl-CoA binding to DmdR derepresses *dmdA-acuI* transcription to stimulate acryloyl-CoA catabolism and DMSP demethylation. Upregulation of the newly identified peroxidase DmdF, and possibly also of DmdE and DinB, counteracts oxidative stress associated with DMSP demethylation. Thus, DmdR, along with DmdR-independent regulators of DMSP cleavage, likely maintains cellular DMSP levels to allow its antistress functions, but accelerates demethylation and catabolism of toxic intermediates at higher DMSP levels. Of note, DmdR appears to control acryloyl-CoA catabolism/detoxification even in abundant marine bacteria lacking *dmdA*, suggesting additional mechanisms. DmdR and DmdEF are widespread in Earth's oceans and important for biogeochemical cycling and climate-active gas production.

**Keywords** DMSP Catabolism; Coordinated Regulation; Transcriptional Regulator; Marine Bacteria
**Subject Categories** Evolution & Ecology; Microbiology, Virology & Host Pathogen Interaction

## Introduction

Marine organisms produce >8 billion tons of dimethylsulfoniopropionate (DMSP) annually (Curson et al, 2017; Galí et al, 2015; Hopkins et al, 2023), with consequences for stress tolerance (Stefels, 2000; Sunda et al, 2002), chemotaxis (Seymour et al, 2010; Wolfe et al, 1997), biogeochemical cycling (Curson et al, 2011b; Kiene et al, 2000; Reisch et al, 2011; Simó et al, 2002) and climate-active gas production (Johnston et al, 2016). Bacterioplankton, particularly Roseobacters, can import and concentrate dissolved DMSP to high millimolar levels (Reisch et al, 2008) allowing it to act as an antistress compound in, e.g., osmoprotection (Curson et al, 2011b; Sun et al, 2012). Many such bacteria can also catabolise DMSP via two possible pathways (Fig. 1A) (Howard et al, 2006; Kiene et al, 2000; Simó et al, 2002). Bacterial DMSP demethylation is initiated by DmdA and can be used for carbon and sulfur (via methanethiol, MeSH) assimilation, and serve as a methyl donor to fuel the methionine cycle (Howard et al, 2006; Reisch et al, 2008; Reisch et al, 2011; Sperfeld et al, 2024). Note, DMSP demethylation has also been reported to cause oxidative stress (Eyice et al, 2018; Schäfer and Eyice 2019; Wang et al, 2022). By contrast, DMSP cleavage is performed by Ddd enzymes in bacteria and fungi (Curson et al, 2008; Curson et al, 2011a; Li et al, 2021; Sun et al, 2016; Todd et al, 2009; Todd et al, 2010; Todd et al, 2011; Todd et al, 2012b; Wang et al, 2023) and Alma-family enzymes in algae and corals (Alcolombri et al, 2015), and generates dimethylsulfide (DMS). This climate-cooling gas has consequences for chemotaxis (Seymour et al, 2010; Wolfe et al, 1997) and climate (Johnston et al, 2016; Mahajan et al, 2015). DMSP cleavage also yields a $C_3$ co-product (acrylate, 3-hydropropionate, or acryloyl-CoA) which can be utilized as a carbon source or to deter predator (Curson et al, 2008; Curson et al, 2011a; Li et al, 2021; Sun et al, 2016; Teng et al, 2021; Todd et al, 2009; Todd et al, 2010; Todd et al, 2011; Todd et al, 2012b; Wang et al, 2023). Bacteria can convert acrylate to cytotoxic acryloyl-CoA through the CoA ligase PrpE, which is detoxified by the acryloyl-CoA reductase AcuI (Fig. 1A) (Reisch et al, 2013).

[1]MOE Key Laboratory of Evolution and Marine Biodiversity, State Key Laboratory of Marine Food Processing and Safety Control, Frontiers Science Center for Deep Ocean Multispheres and Earth System & College of Marine Life Sciences, Ocean University of China, 266003 Qingdao, China. [2]Marine Biotechnology Research Center, State Key Laboratory of Microbial Technology, Shandong University, 266237 Qingdao, China. [3]Laboratory for Marine Biology and Biotechnology, Qingdao Marine Science and Technology Center & Laoshan Laboratory, 266373 Qingdao, China. [4]School of Biological Sciences, University of East Anglia, Norwich Research Park, Norwich NR4 7TJ, UK. [5]Centre for Microbial Interactions, Norwich Research Park, Norwich NR4 7TJ, UK. [6]Quadram Institute Bioscience, Norwich Research Park, Norwick, Norfolk NR4 7UQ, UK. [7]These authors contributed equally: Hui-Hui Fu, Ming-Chen Wang. ✉E-mail: Jonathan.Todd@uea.ac.uk; zhangyz@sdu.edu.cn

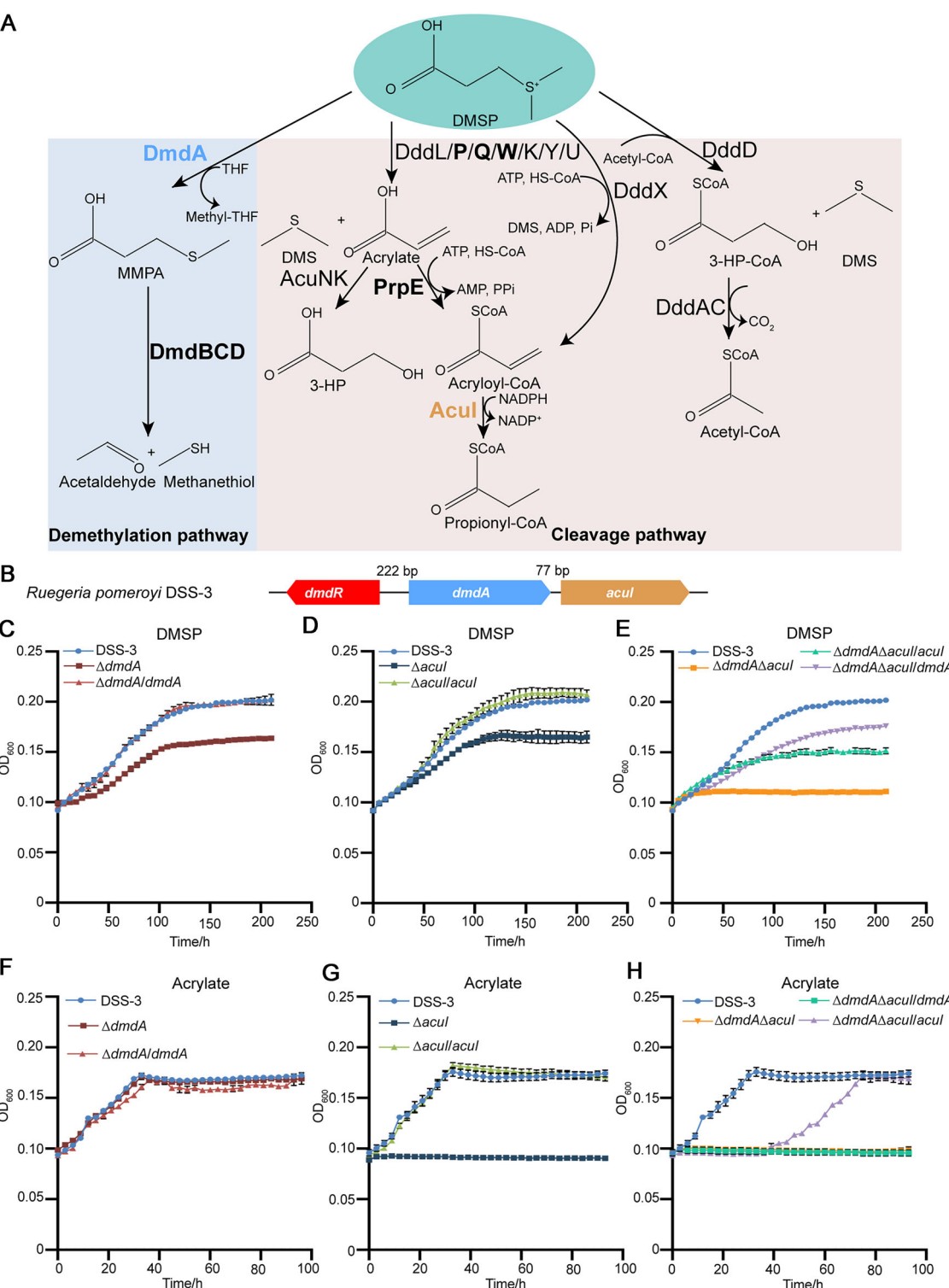

**Figure 1. DMSP catabolic pathways and the importance of *dmdA* and *acuI* in *R. pomeroyi* DSS-3.**

(**A**) The DMSP demethylation (in blue) and cleavage pathways (in salmon). The key enzymes in *R. pomeroyi* DSS-3 are indicated in bold. DMSP dimethylsulfoniopropionate, DMS dimethyl sulfide, MMPA methylercaptopropionate, THF tetrahydrofolate, 3-HP 3-hydroxypropionate. (**B**) Organization of the *R. pomeroyi* DSS-3 *dmdA-acuI* operon and *dmdR*. (**C–H**) Growth of wild-type *R. pomeroyi* DSS-3, Δ*dmdA*, Δ*acuI*, Δ*dmdA*Δ*acuI* and genetically complemented strains in minimal medium with DMSP (6 mM, (**C–E**)) or acrylate (3 mM, (**F–H**)) as sole carbon source, respectively. Data are presented as the mean ± standard deviation ($n \geq 3$). Source data are available online for this figure.

Over 20% of marine bacteria, including most Roseobacters, have the genetic capacity to catabolise DMSP containing *dmdA*, a *ddd* gene or both systems (Cui et al, 2015; Howard et al, 2006; Moran et al, 2003). Indeed, the model Roseobacters *Ruegeria pomeroyi* DSS-3 can both demethylate and cleave DMSP, and use DMSP as a carbon and sulfur source (Fig. 1A) (Moran et al, 2003; Moran et al, 2004; Todd et al, 2011; Todd et al, 2012b). DMSP catabolic enzymes have $K_m$ values in the millimolar range, consistent with a kinetic strategy to maintain high intracellular DMSP levels needed for its physiological function (Brummett et al, 2015; Li et al, 2014; Li et al, 2021; Reisch et al, 2008; Reisch et al, 2011; Shao et al, 2019; Sun et al, 2016; Wang et al, 2015; Wang et al, 2017).

DMSP catabolism is often inducible by DMSP, its catabolites or by both of these (Curson et al, 2011b; Shao et al, 2019; Todd et al, 2011; Todd et al, 2012b; Wang et al, 2017), but it can also be regulated by oxidative stress (Narváez-Barragán et al, 2025). Recently, Gao et al, (2020) showed that DMSP concentration dictates the relative expression of *dddW* and *dmdA* in *R. pomeroyi* DSS-3 to explain the "Switch (Simó et al, 2001)" between the pathways but not the mechanism. A LysR type transcriptional regulator in *R. pomeroyi* DSS-3 controls DMSP-induced *dddW* expression (Todd et al, 2012b). However, the regulator/s controlling *dmdA* and *acuI* were unknown, despite these genes likely being co-transcribed and induced by DMSP and acrylate in many Roseobacters (Todd et al, 2012a). This study elucidates the regulatory mechanism for *dmdA-acuI* expression and provides a mechanistic understanding of the "Switch (Simó et al, 2001)" and how DMSP demethylation and cleavage pathways are coregulated in globally abundant and important marine bacteria. It also identifies novel bacterial *dmd* genes often co-transcribed with *dmdA-acuI* that are essential for DMSP demethylation.

# Results

## *R. pomeroyi* DSS-3 *dmdA* and *acuI* are co-transcribed and essential for growth on DMSP

Initial experiments confirmed that the *R. pomeroyi* DSS-3 *dmdA* and *acuI* genes in DMSP demethylation and cleavage pathways, respectively, formed co-transcribed two gene operon (Fig. 1B, Appendix Fig. S1a). Growth of the Δ*dmdA* and Δ*acuI* strains on DMSP as a sole carbon source was slightly impaired with a reduced maximum yield compared to the wild-type strain ($OD_{max}$ of WT ≈ 0.20, $OD_{max}$ of Δ*dmdA* ≈ 0.16, $OD_{max}$ of Δ*acuI* ≈ 0.16, Fig. 1C,D), but growth was abolished in the Δ*dmdA*Δ*acuI* strain (Fig. 1E). These results indicated that the demethylation or cleavage pathway alone could support growth, but both were required for optimal growth on DMSP. The Δ*dmdA* mutant grew essentially as wild-type on acrylate as sole carbon source (Fig. 1F). By contrast, both Δ*acuI* and Δ*dmdA*Δ*acuI* strains could not grow on acrylate. Growth on acrylate was rescued by the expression of cloned *acuI* (but with an extended lag phase in Δ*dmdA*Δ*acuI*) but not *dmdA* (Fig. 1G,H). These phenotypes were likely due to the toxicity of acrylate added or generated from DMSP cleavage, since Δ*acuI* and Δ*dmdA*Δ*acuI* exhibited significant growth deficits compared to the wild-type even when succinate was provided alongside DMSP/acrylate (Appendix Fig. S1b–d). We noted that cloned *dmdA* no

longer enhanced the growth deficit of the Δ*dmdA*Δ*acuI* strain when succinate was present with DMSP, but the reason for this is unclear. Nevertheless, these data are consistent with previous studies (Curson et al, 2011b; Howard et al, 2006; Reisch et al, 2013) and confirmed that in *R. pomeroyi* DSS-3, unlike in some other Roseobacters (Narváez-Barragán et al, 2025), *dmdA* and *acuI* are essential for DMSP-dependent carbon assimilation, via demethylation and cleavage, respectively.

## DmdR represses *dmdA-acuI* and *dmdR* transcription in a DMSP and acrylate concentration-dependent manner

We proposed the *SPO1912* gene (termed *dmdR*) as the "switch" regulator because it was divergently transcribed directly upstream of *dmdA-acuI* in *R. pomeroyi* DSS-3 (Fig. 1B) and encoded a putative FadR-type transcriptional regulator (Pfam00392) of the GntR superfamily. To test this hypothesis, Δ*dmdR* was generated. Growth yields from DMSP or acrylate, and DMSP-dependent DMS and MeSH production were enhanced in the Δ*dmdR* mutant versus the wild-type, with these enhancements lost in the presence of plasmid-borne *dmdR* (Fig. 2A,B; Appendix Fig. S2a,b). These data implied that *R. pomeroyi* DSS-3 DmdR negatively regulated DMSP catabolism, and that the Δ*dmdR* provided a growth advantage on DMSP or acrylate as sole carbon source, likely through enhanced DmdA and AcuI activity.

Transcription of *dmdA-acuI* was upregulated (<20-fold) by DMSP or acrylate added to 300 μM levels, consistent with the demethylation pathway being activated by micromolar DMSP concentrations in Roseobacters (Sperfeld et al, 2024; Wirth et al, 2020). Upregulation was much more prominent (~66–75 or 95–144-fold, respectively) in response to millimolar levels in the wild-type strain (Appendix Fig. S2c–f), confirming the importance of inducer concentration, as previously demonstrated (Gao et al, 2020; Todd et al, 2012a). Importantly, *dmdA-acuI* transcription was > 50-fold elevated in Δ*dmdR* cells grown in media containing mixed carbon sources (CM) with or without micromolar levels of co-inducers compared to wild-type cells (Fig. 2C; Appendix Fig. S2g). Furthermore, Western blotting with DmdA antibody indicated that DmdA protein levels increased with DMSP and acrylate concentration in wild-type *R. pomeroyi* DSS-3, but were constitutively high in Δ*dmdR* cells (Fig. 2D,E; Appendix Fig. S2h). DMSP and acrylate concentration-dependent transcriptional induction of *dmdR* itself was also observed in wild-type cells (Appendix Fig. S2i,j). In contrast, *dmdR* transcription was enhanced and deregulated in Δ*dmdR* compared to wild-type cells (Appendix Fig. S2i,j). Note, there was still marginal but significant ($P = 0.018$ and $0.006$ for *dmdA*; $P = 0.017$ for *acuI* with increased acrylate levels; $P = 0.012$ and $0.0001$ for *dmdR*) upregulation of *dmdA-acuI* and *dmdR* transcription in Δ*dmdR* with increased DMSP/acrylate levels, implying that there may be other regulators/factors influencing their transcription. Nevertheless, substantial DMSP- and acrylate-dependent regulation of *dmdA-acuI* and *dmdR* was restored by recombinant *dmdR* expressed from a plasmid (Fig. 2C; Appendix Fig. S2g,i,j). These data support DmdR being autoregulatory, consistent with other GntR superfamily regulators (Arya et al, 2021; Dudek et al, 2020), and the DMSP and acrylate responsive transcriptional repressor of *dmdA-acuI* and *dmdR*.

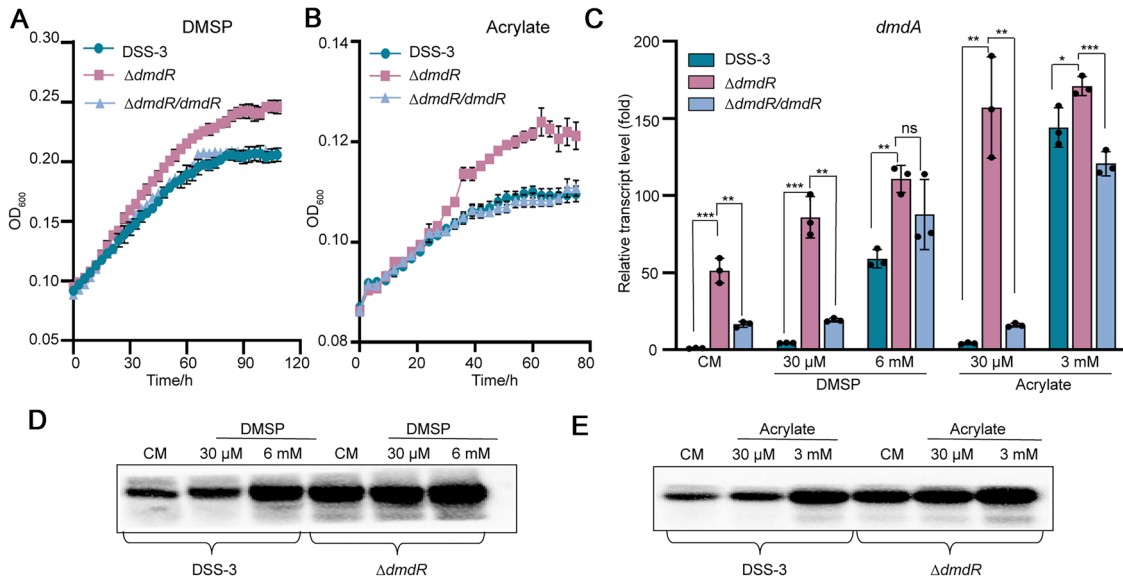

**Figure 2.    DmdR mediated transcriptional regulation of *dmdA-acuI* in *R. pomeroyi* DSS-3.**

(A, B) Growth of wild-type *R. pomeroyi* DSS-3, Δ*dmdR* mutant and genetically complemented (Δ*dmdR/dmdR*) strains in minimal medium with DMSP (6 mM, (A)) or acrylate (3 mM, (B)) as sole carbon source. The $OD_{max}$ values of DSS-3 and Δ*dmdR* grown on DMSP are 0.21 and 0.24, respectively; The $OD_{max}$ values of DSS-3 and Δ*dmdR* grown on acrylate are 0.11 and 0.12, respectively. (C) Relative transcript levels of *dmdA* in wild-type (DSS-3), Δ*dmdR* and Δ*dmdR/dmdR* strains grown with different concentrations of DMSP and acrylate compared to the CM treatment. *P* values in CM treatment: $P_{DSS-3\&\Delta dmdR} = 0.0004$; $P_{\Delta dmdR\&\Delta dmdR/dmdR} = 0.0018$. *P* values in 30 μM DMSP treatment: $P_{DSS-3\&\Delta dmdR} = 0.0004$; $P_{\Delta dmdR\&\Delta dmdR/dmdR} = 0.0010$. *P* values in 6 mM DMSP treatment: $P_{DSS-3\&\Delta dmdR} = 0.0011$; $P_{\Delta dmdR\&\Delta dmdR/dmdR} = 0.1759$. *P* values in 30 μM acrylate treatment: $P_{DSS-3\&\Delta dmdR} = 0.0013$; $P_{\Delta dmdR\&\Delta dmdR/dmdR} = 0.0018$. *P* values in 3 mM acrylate treatment: $P_{DSS-3\&\Delta dmdR} = 0.0305$; $P_{\Delta dmdR\&\Delta dmdR/dmdR} = 0.0009$. (D, E) Western blot analysis of DmdA in wild-type (DSS-3) and Δ*dmdR* cells grown with CM or different concentrations of DMSP (D) or acrylate (E) as indicated. 20.5 μg of protein was analyzed in each lane. Data are presented as the mean ± standard deviation ($n \geq 3$). A two-sided Student's *t* test was used to assess statistically significant differences (***$P < 0.001$; **$P < 0.01$; *$P < 0.05$; ns, $P > 0.05$). All experiments were carried out at least three times. Source data are available online for this figure.

## DmdR binds palindromic *dmd box* sequences at the *dmdA-acuI* promoter

Electrophoretic mobility shift assays (EMSAs) showed significant interaction between purified dimeric DmdR (Appendix Fig. S3a) and the 222 bp *dmdR* and *dmdA* intergenic DNA, termed P1 (Figs. 1B and 3A), which was enhanced with increasing DmdR concentration (Fig. 3B). Two 13 bp direct repeats (TTAAATGT-CAGAC) separated by 2 bp (Fig. 3C), termed *dmd box 1* and *2*, were identified in P1, which matched the palindromic $N_y$GTM-$N_{0-1}$-KACN$_y$ consensus binding motif of FadR family regulators (Suvorova et al, 2015). DNase I footprinting analysis on P1 showed protection of a 32 bp region, spanning *dmd box 1* and *2* (Appendix Fig. S3b). Indeed, this 32 bp DNA, termed P2, exhibited a DmdR concentration-dependent band shift in EMSAs (Fig. 3D). Microscale thermophoresis (MST) showed DmdR had a $K_d$ of 2.63 ± 0.46 μM for P2 (Appendix Fig. S3c).

To establish the importance of *dmd box 1* and *2*, P1 DNA probe variants were constructed (Fig. 3A), lacking *dmd box 1* (P3), *dmd box 2* (P4) or both *dmd box 1* and *2* (P5). DmdR EMSA shifts were detected for probe P3 and P4 (Fig. 3E,F), but importantly not for a control oligonucleotide P5 which lacked a *dmd box* (Appendix Fig. S3d). Consistently, MST analysis detected the interaction of DmdR to P3 and P4, showing ~four-fold higher affinity for P4 containing *dmdA box 1* ($K_d$ 0.26 ± 0.07 μM, Appendix Fig. S3e,f). The respective importance of these *dmd boxes* was not examined further in vivo because they overlapped with the −35 region of *dmdA* (*dmd box 1*) or the −10 regions of *dmdA* and *dmdR*, and the

*dmdR* transcription start site (*dmd box 2*) predicted from 5' rapid amplification of cDNA ends (RACE) analysis (Fig. 3C). These results are consistent with DmdR acting to repress transcription of *dmdA-acuI* and itself by binding to either *dmd box* in the promoter region to prevent transcriptional initiation.

There were no *dmd box* sequences identified with >10/13 bp identity to *dmd box 1* and *2* in intergenic locations or in the same tandem repeated fashion elsewhere in the *R. pomeroyi* DSS-3 genome and megaplasmid (Appendix Table S1). Furthermore, none of the genes adjacent to candidate *dmd boxes* were linked to DMSP or acrylate catabolism. Transcription of genes associated with these candidate *dmd boxes* with ≥8–10/13 bp identity was not enhanced by millimolar DMSP levels in wild-type cells or deregulated in the Δ*dmdR* strain (Appendix Fig. S4a–d). These data implied that *R. pomeroyi* DSS-3 DmdR had a specific role in DMSP/acrylate catabolism.

## The AcuI substrate acryloyl-CoA is the DmdR effector molecule

Studies were conducted to elucidate the DmdR effector molecule. Firstly, a Δ*ddd* mutant (with in-frame deletion of *dddP*, *dddQ* and *dddW* (Todd et al, 2009; Todd et al, 2011; Todd et al, 2012b)) that was devoid of DMSP lyase activity (Appendix Fig. S5a) showed no DMSP-dependent induction of *dmdA* and *acuI* transcription (Fig. 4A; Appendix Fig. S5b). However, DMSP or acrylate addition did not interfere with the DmdR and DNA probe P2 interaction (Appendix Fig. S5c,d), implying they were not the DmdR effector.

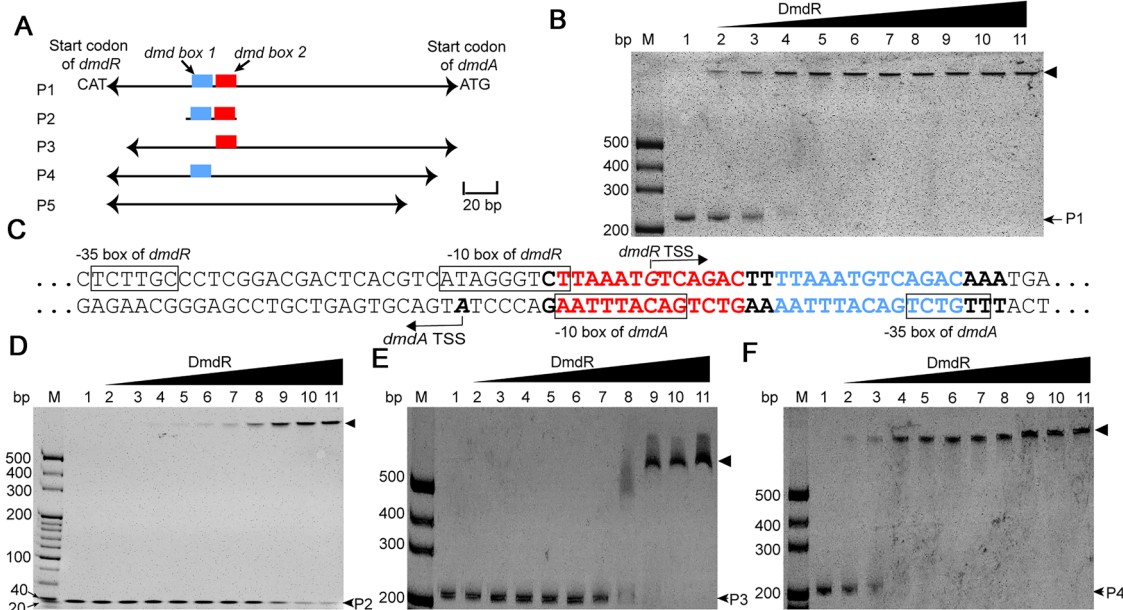

**Figure 3. DmdR binds to *dmd box* sequences in the *dmdR* and *dmdA-acuI* promoters.**

(A) Schematic of DNA probes used in EMSAs. P1: the full-length *dmdR* and *dmdA-acuI* intergenic region; P2: the 32 bp DmdR-protected region revealed by DNase I footprinting (see Appendix Fig. S3b); P3: variant of P1 without *dmd box 1*; P4: variant of P1 without *dmd box 2*; P5: variant of P1 without the 32 bp DmdR-protected region, i.e., without both *dmd boxes*. (B) EMSAs of DmdR titrated against P1 (7 nM). Lane 1, P1; lane 2 to 11 show addition of increasing quantities of DmdR (10, 20, 40, 60, 80, 100, 200, 400, 600, and 800 nM, respectively). M: DNA marker. (C) The partial intergenic space sequence containing the 32 bp DmdR-protected region (in bold). The *dmdA-acuI* and *dmdR* transcription start sites (TSSs) are in bold, italics and indicated with an arrow. −10 and −35 boxes are framed. The DNA sequences of *dmd box 1* and *dmd box 2* were marked in blue and red, respectively. (D–F) EMSAs of DmdR titrated against P2 (237 nM), P3 (7 nM), and P4 (7 nM) as indicated. Lane 1, corresponding DNA probe; Lane 2 to 11 show addition of increasing quantities of DmdR (10, 20, 40, 60, 80, 100, 200, 400, 600, and 800 nM, respectively). The shift band is indicated by black triangle. M: DNA marker. Data presented are from at least three independent experiments. Source data are available online for this figure.

Furthermore, DmdR $K_d$ values for DMSP or acrylate binding were extremely high (~140 mM, Appendix Fig. S5e,f). These data implicated acrylate catabolite/s as the DmdR effector/s.

Acrylate induced *dmdA* transcription was shown to be significantly enhanced above wild-type levels in *R. pomeroyi* DSS-3 Δ*acuI* (Fig. 4B), which cannot catabolise and thus accumulated acryloyl-CoA produced from acrylate by PrpE (propionate-CoA ligase) (Fig. 1A). DmdR bound acryloyl-CoA with a $K_d$ of 9.61 ± 4.22 μM (Fig. 4C). Furthermore, the binding affinity of DmdR to DNA probe P2 was dramatically decreased to millimolar levels in the presence of acryloyl-CoA (Fig. 4D). EMSAs performed in the presence of acrylate further demonstrated acryloyl-CoA-dependent inhibition of DmdR binding target *dmd box* sequences when the acryloyl-CoA ligand was generated in situ with increasing PrpE enzyme levels (Appendix Fig. S5g; Fig. 4E). Controls lacking PrpE (C-1) or with an inactive K588A PrpE mutant (Wang et al, 2017) (C-2) yielded no acryloyl-CoA (Appendix Fig. S5g) and showed binding between DmdR and probe P1 (Fig. 4E). Note, DmdR did not bind the AcuI enzyme product propionyl-CoA (Appendix Fig. S5h) or 3-hydroxypropionate (3-HP, Fig. 1A; Appendix Fig. S5i), a DMSP cleavage pathway product (Todd et al, 2010) and co-inducer of *ddd-acu* gene clusters in other bacteria (Curson et al, 2011a; Curson et al, 2011b). Moreover, these other potential effectors did not impact DmdR binding to DNA probe P2 (Appendix Fig. S5j,k). Collectively, these data indicated that acryloyl-CoA attenuated the DmdR interaction with *dmd boxes* and was the DmdR effector.

## The proposed DmdR regulatory mechanism

Prediction of the DmdR structure was combined with mutagenesis of key residues to study its regulatory mechanism. DmdR domain architecture was typical of FadR family regulators, with an N-terminal DNA-binding domain and a C-terminal effector-binding and oligomerization (E-O) domain consisting of 7 helices (Suvorova et al, 2015) (Appendix Fig. S6a). Alphafold 2 predicted DmdR to form a domain-swapped dimer through the E-O domain, consistent with other FadR family members (Horne et al, 2021; Resch et al, 2010) (Appendix Fig. S6b). Substitution mutations in Gln85, Glu136, Asp163, Ser171 and Glu179 residues in the E-O domain, predicted to be the acryloyl-CoA binding site (Appendix Fig. S6c), either showed no acryloyl-CoA binding or had binding affinities in the millimolar range (Gln85Ala and Glu136Ala) compared to the micromolar levels seen with the native DmdR protein (Appendix Fig. S7a–e; Fig. 4C). These site-directed variants displayed circular dichroism (CD) spectra profiles similar to native DmdR (Appendix Fig. S7f) and also formed dimers in solution (Appendix Fig. S8). These data were consistent with reduced acryloyl-CoA binding ability being due to residue replacement rather than global secondary structure or oligomeric status changes. Furthermore, *dmdA* transcript levels were substantially lower in Δ*dmdR* cells containing a plasmid-borne *dmdR* with these site-directed mutations compared to wild-type *dmdR* (Appendix Fig. S6d). Of these site-directed mutations, D163A, S171A, and E179A exhibited lower *dmdA* transcript levels compared to Q85A

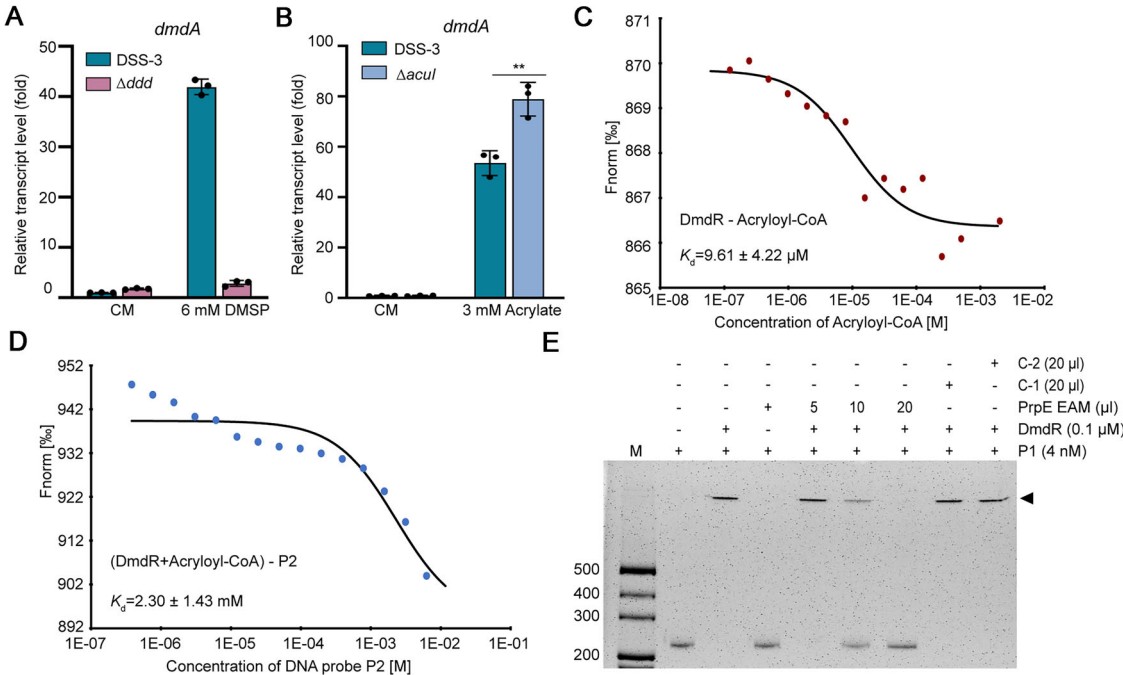

**Figure 4. Acryloyl-CoA is the DmdR effector.**

(A) Relative transcript levels of *dmdA* in wild-type DSS-3 and Δ*ddd* strains grown with CM or DMSP (6 mM). (B) Relative transcript levels of *dmdA* in wild-type DSS-3 and Δ*acuI* strains grown with CM or acrylate (3 mM). $P = 0.006$. (C) MST analysis of DmdR binding to acryloyl-CoA. The acryloyl-CoA was titrated from 1.22 μM to 2 mM. DmdR had a $K_d$ of 9.61 ± 4.22 μM for acryloyl-CoA. (D) MST analysis of DmdR binding to DNA probe P2 in the presence of acryloyl-CoA (1.25 μM). The DNA probe P2 was titrated from 381 nM to 12.5 mM. The $K_d$ of DmdR for P2 dramatically increased in the presence of acryloyl-CoA to 2.30 ± 1.43 mM. (E) Competing EMSA analysis of interactions between DmdR and DNA probe P1 (0.3 nM) in the presence of increasing amount of enzymatic assay mixture (EAM) of PrpE. C-1 and C-2, incubation mixture without recombinant PrpE and with K588A variant of PrpE, respectively. The shift band is indicated by black triangle. M: DNA marker. Data are presented as the mean ± standard deviation ($n \geq 3$). A two-sided Student's *t* test was used to assess statistically significant differences (**$P < 0.01$). Data presented are at least three independent experiments. Source data are available online for this figure.

and E136A, implying that D163, S171, and E179 were likely more critical for the interaction between DmdR and its effector. This effector binding site was predicted at the interface of the two monomers, implying that the DmdR dimer bound one acryloyl-CoA molecule, as with e.g., the GntR type transcriptional repressor NanR (Horne et al, 2021).

We propose that DmdR represses *dmdA-acuI* transcription by binding to either *dmd box* in its promoter region in the absence of inducer, i.e., when external and cellular DMSP levels are low. In addition, *dddP*, *dddQ* and *dddW* transcription, which were independent of DmdR control (Appendix Fig. S9a–c) and not acrylate inducible (consistent with previous studies (Todd et al, 2009; Todd et al, 2011; Todd et al, 2012b)), remain at low levels (Appendix Fig. S9d). Furthermore, given the widely recognized millimolar $K_m$ values of DMSP lyases (Brummett et al, 2015; Li et al, 2014; Wang et al, 2015), these enzymes are unlikely to liberate large quantities of acrylate or its catabolites when intracellular DMSP is below millimolar levels. Thus, when dissolved DMSP is sparse and intracellular levels are low, DMSP catabolism is switched off, favoring intracellular accumulation (Fig. 5A). Exposure to higher dissolved DMSP concentrations allows DMSP to accumulate intracellularly to millimolar levels (Reisch et al, 2008). Under these conditions, DMSP lyase genes are induced (Appendix Fig. S9d). Note, *dddW* is likely key in this since it has the lowest threshold for DMSP induced transcription, probably due to its

LysR-family regulator (Landa et al, 2017; Todd et al, 2012b). Enhanced DMSP cleavage accelerates acrylate and subsequently acryloyl-CoA production, which are potentially toxic to cells (Todd et al, 2012a; Wang et al, 2017). Note, *prpE* transcription showed DmdR-independent DMSP induction, like *dddP* and *dddW*, consistent with more acryloyl-CoA being produced with increased DMSP concentration (Appendix Fig. S9a–e). Binding of the acryloyl-CoA effector to the DmdR E-O domain likely results in a conformational change that attenuates the DmdR-*dmd box* interaction and causes upregulation of *dmdA-acuI* transcription. Notably, similar allosteric mechanisms are used by other GntR family regulators (Arya et al, 2021; Dudek et al, 2020; Horne et al, 2021; Resch et al, 2010). Upregulation of *acuI*/AcuI results in swift catabolism of acryloyl-CoA to non-toxic propionyl-CoA, which enters central metabolism. Meanwhile, the demethylation pathway is also upregulated to accelerate DMSP catabolism (Fig. 5B).

## DmdR and its regulatory mechanism is widespread in Roseobacters

The divergon arrangement of *dmdR* and its *dmdA-acuI* regulon is typical of GntR regulators (Suvorova et al, 2015). To study the ubiquity of DmdR regulation its conservation and gene synteny was investigated in bacteria, initially Roseobacters. Importantly, in 33 of 53 Roseobacters genomes the gene for bona fide DmdA, was in a

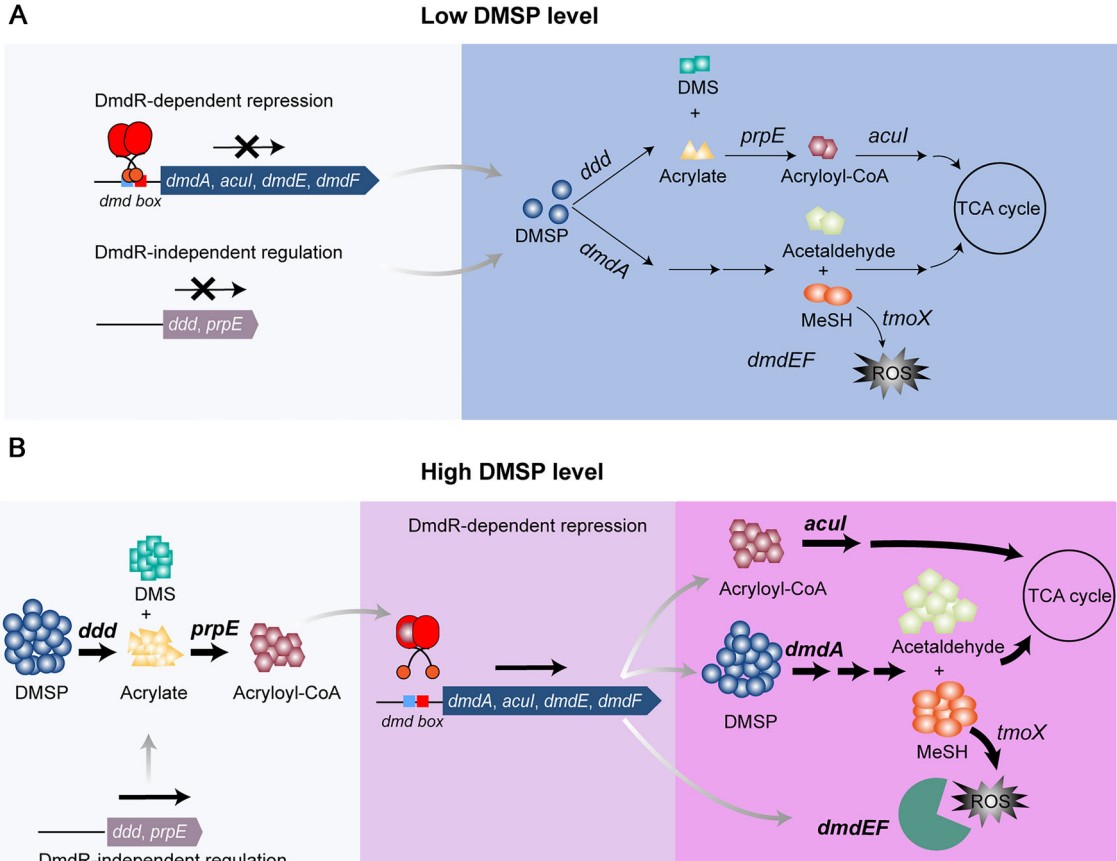

**Figure 5. DmdR mediated regulation of DMSP catabolism.**

(A) When DMSP concentration is low, the DmdR dimer binds to *dmd box* sequences in the promoter region of *dmdA-acuI* (including *dmdEF* in many cases), resulting in repression of DMSP demethylation, oxidative stress protection and downstream DMSP cleavage (catalyzed by enzyme AcuI), ensuring conditions that favor its intracellular accumulation. (B) When DMSP concentration increases, *ddd* gene expression is increased, leading to enhanced DMSP cleavage and accumulation of the DmdR effector acryloyl-CoA through the action of PrpE. DmdR binds to its acryloyl-CoA effector. Holo-DmdR with lower affinity to *dmd boxes* dissociates from the *dmdA-acuI* promoter derepressing transcription. This upregulates expression of the demethylation pathway and the AcuI enzyme of the DMSP cleavage pathway, that detoxifies acryloyl-CoA and enables DMSP's use as a carbon and sulfur source. Enhanced DMSP demethylation results in reactive oxygen species (ROS) production and oxidative stress, which can be alleviated via the action of the peroxidase DmdF and, potentially, DmdE and DinB, whose expression is also derepressed.

predicted operon with *acuI* alone or with *acuI* plus other conserved genes not currently linked to DMSP or acrylate catabolism (Appendix Fig. S10). Note, bona fide DmdA sequences were those phylogenetically clustering with characterised DmdA proteins and not the non-functional *R. pomeroyi* DSS-3 DmdA-like protein (SPO1648) (Howard et al, 2006). Of these bacteria, all except Rhodobacterales bacterium HTCC2150 had a divergently transcribed candidate *dmdR* immediately upstream of an operon containing *dmdA-acuI* (Appendix Fig. S10). These DmdR candidates shared > 66% protein sequence identity to *R. pomeroyi* DSS-3 DmdR (Appendix Table S2). Furthermore, these Roseobacters *dmdR* divergon promoter regions contained two palindromic *dmd box* sequences ($N_yGTM-N_{0-1}-KACN_y$), separated by varying lengths of nucleotides (Appendix Fig. S11a).

We further examined DmdR-mediated regulation in *Ruegeria lacuscaerulensis* ITI-1157 that demethylates and cleaves (with *dddP* and *dddQ* genes) DMSP (Curson et al, 2011b; Wirth et al, 2020). In this Roseobacter, *dmdA* and *acuI* were co-transcribed at the distal end of an operon with three genes of unknown function (see below,

Appendix Figs. S10 and S11b). Transcription of *R. lacuscaerulensis* ITI-1157 *dmdA-acuI* and the three additional genes was induced by DMSP, deregulated in a Δ*dmdR* background, but restored by *dmdR* expressed in trans (Appendix Fig. S11c–g). Furthermore, DMSP-dependent DMS and MeSH production was also enhanced in the Δ*dmdR* compared to wild-type (Appendix Fig. S11h,i). These data imply that DmdR, via the sensing of acryloyl-CoA, is the central regulator governing DMSP demethylation and downstream DMSP cleavage pathways in most Roseobacter members that contain these two catabolic pathways.

In subsequent bioinformatics, the NCBI genome database was screened for bacteria in which *dmdR*, *dmdA* and *acuI* were clustered together (i.e., within a 10 open reading frame cluster). Those bacteria containing *dmdR* close to *dmdA* and *acuI* were found to be exclusively Alphaproteobacteria, mainly Rhodobacterales, and some Hyphomicrobiales, but not SAR11 bacteria which all lacked *dmdR* (Appendix Fig. S12). In contrast, bacteria harboring "non-clustered" *dmdR* were far more diverse, including Actinomycetes, Gammaproteobacteria, Betaproteobacteria and some

                                                                    

Alphaproteobacteria (Appendix Fig. S12). Many of these "non-clustered" bacteria with *dmdR* lacked *dmdA*, particularly Oceanospirillales known to cleave DMSP for carbon assimilation (Curson et al, 2011b; Liu et al, 2022), but they did contain *dmdR* divergent to *acuI* (Appendix Fig. S12). Indeed, DmdR homologs from the Oceanospirillales *Neptunnibacter halophilus* LMG 25378 and *Marinobacterium sedimentorum* SO208 interacted with the *dmdR-acuI* intergenic region (Appendix Fig. S13). There were no instances where *dmdR* was divergent to only *dmdA*. These data imply that DmdR was originally an acryloyl-CoA responsive regulator of *acuI*, see discussion.

## DmdR-regulated genes *dmdEF* are essential for DMSP demethylation and associated with oxidative stress protection

Given the operonic structure and DmdR-regulated expression of the three genes upstream of *dmdA-acuI* in *R. lacuscaerulensis* ITI-1157 and other Roseobacters (Appendix Fig. S10), they may be involved in the catabolism of DMSP, acrylate or both molecules. The first two genes in this operon (SAMN05444404_0805-0806, now termed *dmdEF*) encoded 'DmdE' a predicted metal-binding integral membrane protein (DUF2182) with a signal peptide (predicted by PSORTb v3.0.2) (Yu et al, 2010) and 'DmdF' a DUF1326 domain-containing protein, both of which had no known function. The third gene (SAMN05444404_0807) encoded a damage inducible (DinB) family protein, induced in response to environmental stresses e.g., oxidative stress, in diverse bacteria (Jarosz et al, 2007; Mckenzie et al, 2003). *dmdE* and *dmdF* were mostly in alphaproteobacterial Rhodobacterales, always adjacent, and often with *dinB*-like genes, implying their products function together, have related roles, or possess both these properties (Appendix Figs. S10 and S12; Appendix Table S3). Furthermore, Rhodobacterales *dmdEF* were always situated close to *dmdR* in an operonic structure with *dmdA* and *acuI* or *acuI* only, suggesting a function linked to DMSP or acrylate catabolism (Appendix Figs. S14 and S15; Appendix Table S3). *dmdEF*-like genes were also present in Betaproteobacteria, Gammaproteobacteria, and Actinomycetes, but were phylogenetically distinct to those linked with DMSP/acrylate catabolic genes, and these were not located in the vicinity of known DMSP or acrylate catabolic genes (Appendix Figs. S14 and S15; Appendix Table S3). Note, *dmdE* and *dmdF* were absent in the genomes of *R. pomeroyi* DSS-3 and SAR11 bacteria.

*R. lacuscaerulensis* ITI-1157 Δ*dmdE_{Rl}* and Δ*dmdF_{Rl}* mutants failed to grow on DMSP, and Δ*dinB_{Rl}* exhibited impaired growth on DMSP (Fig. 6A). Whilst the growth of Δ*dmdE_{Rl}* on DMSP was fully restored by plasmid-borne *dmdE_{Rl}*, Δ*dmdF_{Rl}* was only partially restored by *dmdEF_{Rl}* expressed in trans. This data further implied that DmdEF had related function or that their regulated expression was important. Note, Δ*dmdE_{Rl}* and Δ*dmdF_{Rl}* also exhibited growth defects in minimal medium with 1 mM succinate plus DMSP (Fig. 6B). Interestingly, Δ*dmdE_{Rl}*, Δ*dmdF_{Rl}* (albeit with an extended ~75 h lag phase) and Δ*dinB_{Rl}* could grow on acrylate (Fig. 6C), implying that their role was more critical to DMSP than acrylate catabolism. Moreover, mutations in *dddP_{Rl}* and *dddQ_{Rl}* (the only two DMSP lyases in *R. lacuscaerulensis* ITI-1157 (Wirth et al, 2020)) in the Δ*dmdE_{Rl}* background did not improve its growth on DMSP (Fig. 6D). However, deletion of *dmdA_{Rl}* partially restored

the growth defects of Δ*dmdE_{Rl}* (Fig. 6D), implying that *dmdE_{Rl}* and probably *dmdF_{Rl}* were essential for DMSP demethylation in *R. lacuscaerulensis* ITI-1157.

As a predicted membrane protein, DmdE may be a DMSP transporter. However, this possibility was discounted because the wild-type, Δ*dmdE_{Rl}* and the complemented Δ*dmdE_{Rl}* cells contained similar DMSP levels after incubation with DMSP (Fig. 6E). Furthermore, when DmdEF_{Rl} was expressed in *E. coli*, no significant DMSP import was observed above the background levels, which were far below that seen with the bona fide DMSP transporter DddT from *Halomonas* (Todd et al, 2010) (Fig. 6F). Further work is required to determine the function of DmdE in DMSP catabolism.

Purified DmdF was shown to have peroxidase enzyme activity (Fig. 6G), consuming hydrogen peroxide ($H_2O_2$) and liberating oxygen (Fig. 6H). Given $H_2O_2$ is generated by the demethylation pathway (Eyice et al, 2018; Schäfer and Eyice 2019; Wang et al, 2022), it would make sense for DmdF expression to be upregulated with DinB (previously shown to be involved in oxidative stress protection (Jarosz et al, 2007; Mckenzie et al, 2003)) during DMSP demethylation. Indeed, intracellular ROS levels were elevated in Δ*dmdF* strain compared to wild-type when grown on DMSP (Appendix Fig. S16). Note, we cannot exclude that DmdF may work on other cytotoxic compounds derived from $H_2O_2$, which warrants further study.

## The influence of oxidative stress on *dmdR* and *dmdA* transcription

Given the association of DMSP demethylation (Eyice et al, 2018; Schäfer and Eyice 2019; Wang et al, 2022) and DmdR to oxidative stress and its amelioration, respectively, the impacts of $H_2O_2$ addition on *dmdR* and *dmdA* transcription were examined in *R. lacuscaerulensis* ITI-1157 and *R. pomeroyi* DSS-3. RT-qPCR analysis revealed that $H_2O_2$ addition decreased both *dmdR* and *dmdA* transcription to similar extents (Appendix Fig. S17). Moreover, the same decreased transcript profile was seen for *R. pomeroyi* DSS-3 *dmmA*, which encodes a dimethylamine monooxygenase entirely unrelated to DMSP catabolism. These data imply that oxidative stress does not specifically regulate *dmdR* or *dmdA*, and that *dmdR* responds to DMSP rather than oxidative stress per se. This may be different in other Roseobacters and further work is required to evaluate the impacts of oxidative stress on DMSP catabolism.

## DmdR, DmdE and DmdF are environmentally important

To infer the importance of DmdR and its regulon, the abundance and expression of *dmdR, dmdE* and *dmdF* were examined in marine multiomics datasets. *dmdR, dmdE* and *dmdF* and their transcripts were detected in all *Tara* Oceans metagenomes and metatranscriptomes analyzed, as were the ubiquitous *dmdA* and *acuI* genes (Landa et al, 2019) (Fig. 7A,B; Appendix Table S4). Up to 13.9% of marine prokaryotes (~2.3% average) were predicted to contain *dmdR* and its transcripts were evenly distributed between surface (SRF) and deep chlorophyll maximum (DCM) samples, but were significantly more abundant in the mesopelagic (MES) water layer (Fig. 7A,B; Appendix Table S4). In contrast, 2.3% and 2% of marine prokaryotes contained *dmdE* and *dmdF* (Fig. 7A). When applied to

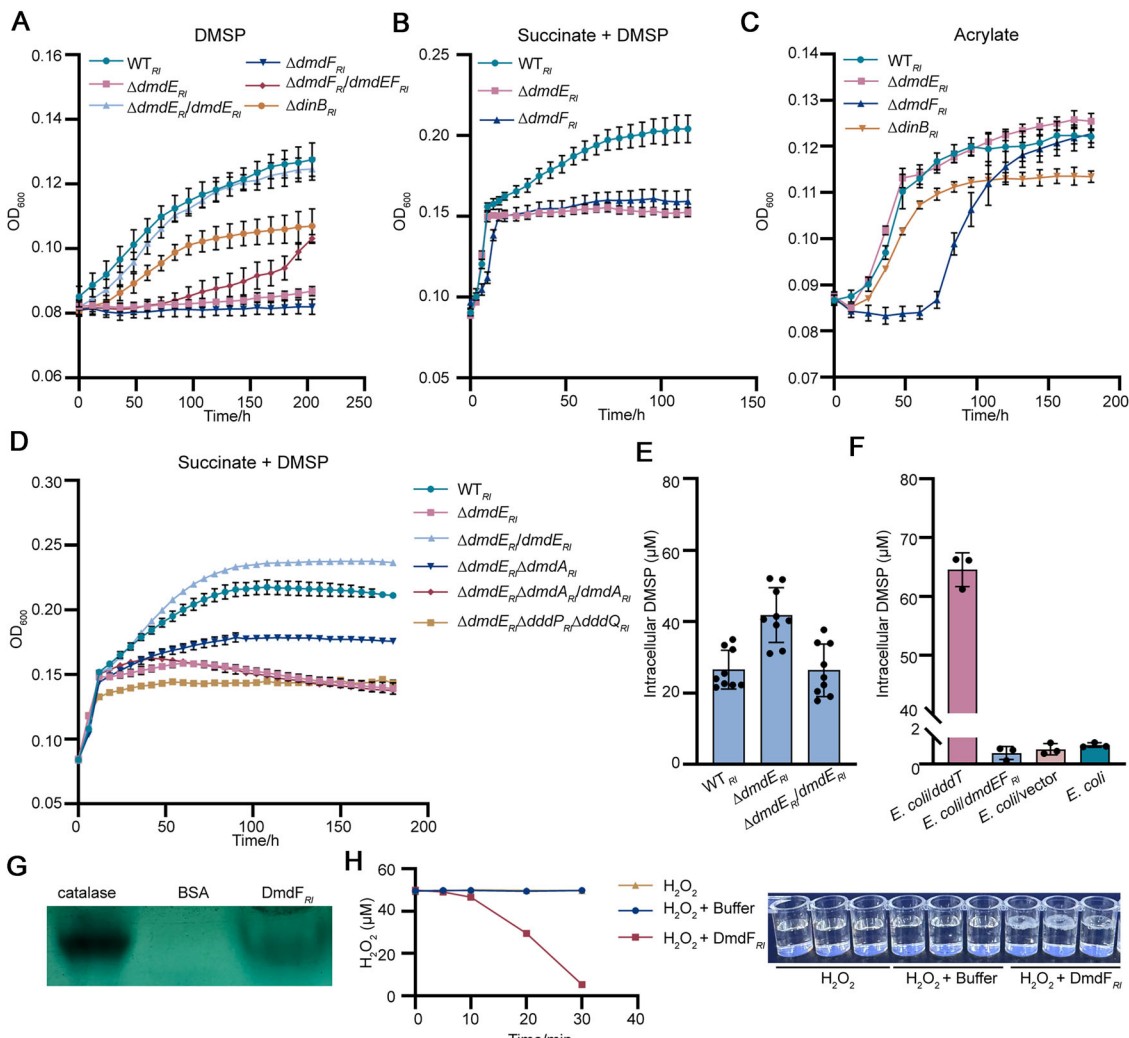

**Figure 6. DmdE and DmdF are important in DMSP catabolism.**

(A) Growth of *R. lacuscaerulensis* ITI-1157 wild-type, $\Delta dmdE_{RI}$, $\Delta dmdF_{RI}$ and $\Delta dinB_{RI}$ and genetically complemented strains ($\Delta dmdE_{RI}/dmdE_{RI}$, $\Delta dmdF_{RI}/dmdEF_{RI}$) in minimal medium with DMSP (5 mM) as sole carbon source. (B) Growth of *R. lacuscaerulensis* ITI-1157 wild-type, $\Delta dmdE_{RI}$ and $\Delta dmdF_{RI}$ strains in minimal medium with 1 mM succinate and 5 mM DMSP as carbon sources. (C) Growth of *R. lacuscaerulensis* ITI-1157 wild-type, $\Delta dmdE_{RI}$, $\Delta dmdF_{RI}$ and $\Delta dinB_{RI}$ in minimal medium with acrylate (3 mM) as sole carbon source. (D) Growth of *R. lacuscaerulensis* ITI-1157 wild-type, $\Delta dmdE_{RI}$, $\Delta dmdE_{RI}/dmdE_{RI}$, $\Delta dmdE_{RI}\Delta dmdA_{RI}$, $\Delta dmdE_{RI}\Delta dmdA_{RI}/dmdA_{RI}$, and $\Delta dmdE_{RI}\Delta dddP_{RI}\Delta dddQ_{RI}$ in minimal medium with 1 mM succinate and 5 mM DMSP as carbon sources. (E) Intracellular DMSP concentrations of *R. lacuscaerulensis* ITI-1157 wild-type, $\Delta dmdE_{RI}$ mutant and genetically complemented ($\Delta dmdE_{RI}/dmdE_{RI}$) strains in minimal medium with DMSP (5 mM). (F) Intracellular DMSP concentration with *E. coli* WM3064 expressing pHGE-P$_{tac}$-$dmdEF_{RI}$ or empty vector. *E. coli* WM3064 harboring reported DMSP transporter $dddT$ cloned in the same vector and *E. coli* WM3064 per se were used as positive and negative control, respectively. (G) In-gel peroxidase activities of purified DmdF. Catalase and BSA (Bovine Serum Albumin) were used as a positive control and a negative control, respectively. 440 μg protein was loaded in each lane. (H) Consumption of $H_2O_2$ by purified DmdF. Buffer, inoculated with protein buffer of DmdF. Data are presented as the mean ± standard deviation ($n \geq 3$). All experiments were carried out at least three times. Source data are available online for this figure.

previous estimates of bacterial cell numbers in seawater (Whitman et al, 1998), this equates to ~$16.9 \times 10^4$, $16.9 \times 10^4$ or $14.7 \times 10^4$ bacteria per ml seawater containing *dmdR*, *dmdE* and *dmdF*, respectively. As expected, *dmdA* and *acuI* (predicted in 26.9% and 93% of marine bacteria, respectively) and their transcripts were far more abundant than for *dmdR*, *dmdE* and *dmdF*. *dmdA*, *dmdE* and *dmdF* sequences and transcripts were more concentrated in photic SRF waters, consistent with previous studies (Landa et al, 2019). However, *acuI* was evenly distributed in the SRF and DCM but was more abundant in MES water bodies, as seen for *dmdR* (Fig. 7A,B).

Taxonomic assignment from *Tara* Oceans datasets also showed that all Pelagibacterales (SAR11) lacked *dmdR*, *dmdE* and *dmdF* (Fig. 7C). This in part accounted for their lower abundance/transcription compared to *dmdA* and *acuI*, since up to 30% and 20% of *acuI* and >50% and >40% of *dmdA* genes and transcripts, respectively, were from SAR 11 genomes (Fig. 7C). Note, SAR11 genomes were also previously reported to host ~80% of marine bacterial DMSP catabolic genes (*dmdA*, *dddK* and *dddP*) (Landa et al, 2019). Consistent with the NCBI analysis (Appendix Fig. S12), *dmdR* was predominantly in Alphaproteobacteria and

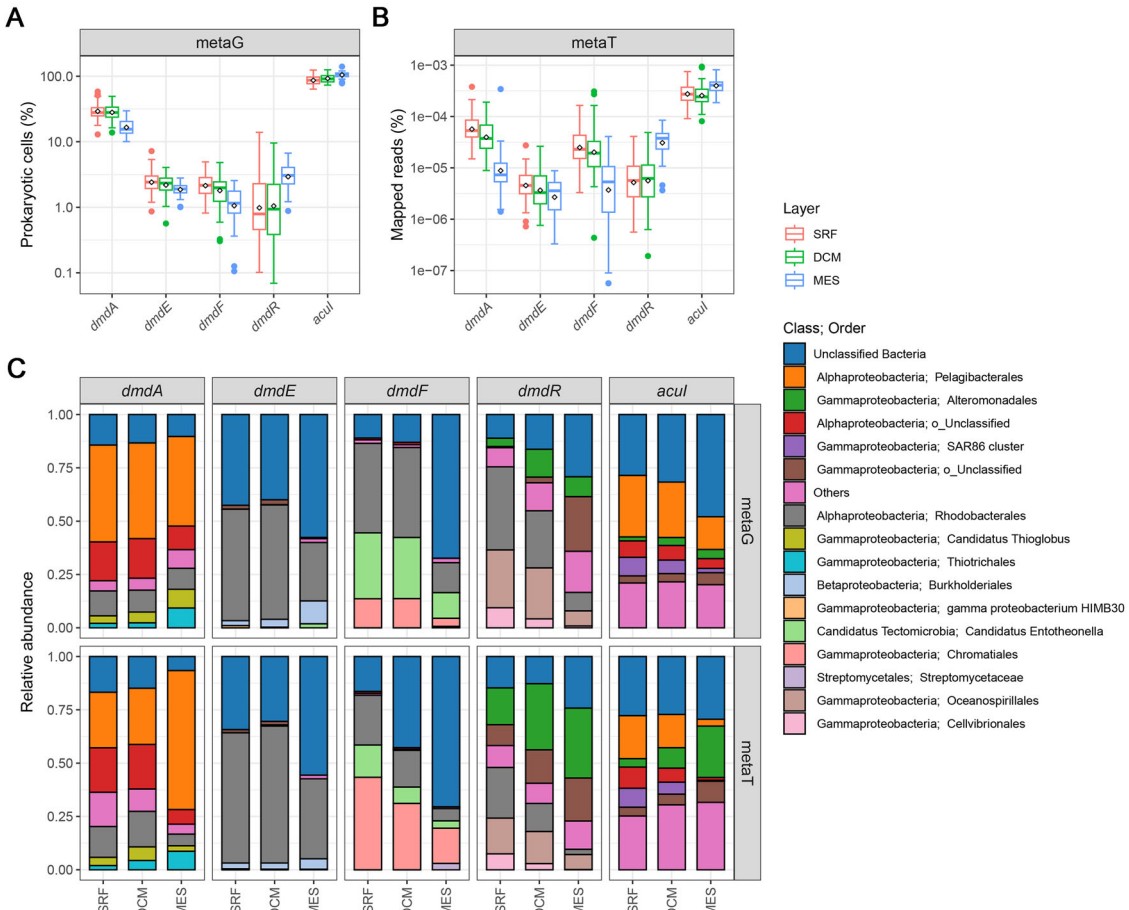

**Figure 7. Distribution of *dmdA*, *dmdE*, *dmdF*, *dmdR* and *acuI* genes in *Tara* Oceans bacterioplankton datasets OM-RGC-v2 (0.22–3 μm).**

(A) Relative abundance of *dmdA*, *dmdE*, *dmdF*, *dmdR* and *acuI* genes in 174 metagenomes from SRF (*n* = 83), DCM (*n* = 53) and MES (*n* = 38) water layers. (B) Relative expression of *dmdA*, *dmdE*, *dmdF*, *dmdR* and *acuI* genes in 178 metatranscriptomes from SRF (*n* = 103), DCM (*n* = 49) and MES (*n* = 26) water layers. (C) Taxonomic assignment of *dmdA*, *dmdE*, *dmdF*, *dmdR* and *acuI* genes in different water layers based on metagenomes and metatranscriptomes. Boxplots show mean (white rhombus), median (center line), upper and lower quartiles (box limits), the interquartile range (whiskers) and outliers (dots). metaG metagenome; metaT metatranscriptome.

Gammaproteobacteria, especially Roseobacters (Rhodobacterales) and Oceanospirillales, respectively (Appendix Fig. S12). In these two major groups of DMSP catabolic bacteria (Curson et al, 2011b; Liu et al, 2022), *dmdR* clustered with *acuI* alone or with *dmdA* and *acuI*, implying that DmdR controls either acrylate or DMSP plus acrylate catabolism, respectively. *dmdE* and *dmdF* were also abundant in Roseobacters and some unclassified bacteria. Furthermore, *dmdF* was also in phototrophic gammaproteobacterial Chromatiales and sponge symbiotic Candidatus Entotheonella bacteria.

## Discussion

This work on DmdR provides an explanation of the "switch" mechanism that controls flux through the competing DMSP demethylation and cleavage pathways in most Roseobacters that are abundant in Earth's oceans and its margins. DMSP demethylation is thought to be the dominant marine catabolic pathway (Howard et al, 2006; Kiene et al, 2000; Venter et al, 2004). However,

transcriptional initiation of DMSP demethylation in Roseobacters likely requires intracellular DMSP levels to be considerable (in the millimolar range) and DMSP cleavage which yields the climate-cooling gas DMS, but more importantly here, acrylate. The further charging of acrylate with a CoA moiety via PrpE, whose gene transcription is also induced by DMSP, generates the DmdR effector molecule acryloyl-CoA. The DmdR holodimer derepresses *dmdA* and *acuI* transcription to stimulate DMSP demethylation, and the detoxification of acryloyl-CoA and demethylation-generated $H_2O_2$ (via novel DmdF and DinB proteins). The eloquent design and balance of this DmdR-mediated system potentially allow the accumulation of intracellular DMSP to levels where it can deliver antistress functions (Carrión et al, 2023b) and the low-level cleavage of DMSP to generate DMS e.g. as a signaling molecule. Importantly, it also allows responsiveness to threshold acryloyl-CoA levels resulting from DMSP cleavage when DMSP levels increase. This facilitates acryloyl-CoA detoxification and the utilization of DMSP as a carbon source via downstream enzymes of the DMSP cleavage pathway, and as a carbon and sulfur source via demethylation. Thus, roseobacterial DmdR is a regulator of

global importance in marine sulfur cycling and climate active gas production.

The above model for DmdR function implies that flux through DMSP cleavage should initially be greater than demethylation in marine Roseobacters. This is quite the opposite to what is reported in previous studies where demethylation dominates DMSP degradation in situ (Howard et al, 2006; Kiene et al, 2000; Venter et al, 2004). Further work on model organisms and diverse environmental samples in carefully controlled settings, which are closer to physiological conditions, are essential to better understand DmdR's function in the environment. Nevertheless, there is no doubt DMSP cleavage enhances DMSP demethylation activity.

This study focused on DmdR-dependent regulation of DMSP catabolism in response to substrate availability but did not consider the role that the sulfur status of the cell or environment plays. This potentially critical factor should also be considered in future studies, as should the identification of unknown regulators that exert DMSP-induced *dddP* and *prpE* transcription in, e.g., *R. pomeroyi* DSS-3.

Ubiquitous SAR11 bacteria both cleave and demethylate DMSP (Carrión et al, 2023a) yet lack DmdR. This implies SAR11 either have a distinct "switch regulator" or that they may lack a transcriptional regulatory system for DMSP demethylation. Such lack of transcriptional regulatory systems is thought to be common in SAR11 (Sun et al, 2016). However, note, transcription of the DMSP lyase gene *dddK* was recently shown to be upregulated by DMSP availability in SAR11 (Carrión et al, 2023a). Further research is required to elucidate the regulatory controls of DMSP catabolism in SAR11 bacteria.

In other abundant marine bacteria, e.g., *Oceanospirillales*, DmdR is predicted to only control acrylate catabolism. It is an appealing hypothesis that the "switch" mechanism in the Roseobacters evolved through fortuitous insertion of *dmdA* into an existing DmdR regulatory system with *acuI*, which allowed the coordinated control of DMSP catabolism and detoxification of DMSP metabolites.

This study also reveals a key requirement to co-regulate oxidative stress protection genes/proteins with DMSP demethylation in abundant Roseobacters and, potentially, many other important marine bacteria via novel systems. In DMSP demethylating bacteria like *R. lacuscaerulensis*, this role is likely undertaken by the novel DmdF peroxidase gene product, identified here, whose expression is upregulated by DmdR in the presence of acryloyl-CoA with DmdA and AcuI. A DinB protein associated with environmental/oxidative stress in diverse bacteria (Jarosz et al, 2007; McKenzie et al, 2003), is also often genetically linked and controlled by DmdR, and is thus likely involved in oxidative stress tolerance associated with DMSP demethylation. This study also identified DmdE, a DmdR-regulated membrane protein of unknown function critical to DMSP demethylation. Like DmdR, these three novel genes were abundant in marine environments and are central to the most prevalent marine DMSP catabolic pathway, and thus global sulfur and carbon cycling. Finally, since many DMSP demethylating bacteria lack DmdE and DmdF, we propose they, e.g. SAR11 and *R. pomeroyi* DSS-3, contain distinct peroxidases that may be upregulated with DMSP demethylation. A potential candidate would be the catalase gene *katG* (Ieva et al, 2008) whose expression in *R. pomeroyi* DSS-3 is known to be induced by DMSP (Wang et al, 2022). Furthermore, *dmdA* and

*katG* are reported to be co-expressed in environmental datasets (Varaljay et al, 2015).

# Methods

## Reagents and tools table

| Reagent/resource | Reference or source | Identifier or catalog number |
|---|---|---|
| **Experimental models** | | |
| *E. coli* WM3064 | W. Metcalf, UIUC | N/A |
| *E. coli* BL21(DE3) | Novagen | N/A |
| *E. coli* DH5α | Novagen | N/A |
| *R. pomeroyi* DSS-3 | Laboratory preservation | N/A |
| *R. lacuscaerulensis* ITI-1157 | Laboratory preservation | N/A |
| Δ*dmdA* | This study | N/A |
| Δ*acuI* | This study | N/A |
| Δ*dmdA*Δ*acuI* | This study | N/A |
| Δ*dmdR* | This study | N/A |
| Δ*ddd* | This study | N/A |
| Δ*dmdA/dmdA* | This study | N/A |
| Δ*acuI/acuI* | This study | N/A |
| Δ*dmdA*Δ*acuI/dmdA* | This study | N/A |
| Δ*dmdA*Δ*acuI/acuI* | This study | N/A |
| Δ*dmdR/dmdR* | This study | N/A |
| Δ*dmdR*<sub>RI</sub> | This study | N/A |
| Δ*dmdR*<sub>RI</sub>/*dmdR*<sub>RI</sub> | This study | N/A |
| Δ*dmdE*<sub>RI</sub> | This study | N/A |
| Δ*dmdF*<sub>RI</sub> | This study | N/A |
| Δ*dinB*<sub>RI</sub> | This study | N/A |
| Δ*dmdE*<sub>RI</sub>/*dmdE*<sub>RI</sub> | This study | N/A |
| Δ*dmdF*<sub>RI</sub>/*dmdEF*<sub>RI</sub> | This study | N/A |
| Δ*dmdE*<sub>RI</sub>Δ*dmdA*<sub>RI</sub> | This study | N/A |
| Δ*dmdE*<sub>RI</sub>Δ*dmdA*<sub>RI</sub>/*dmdA*<sub>RI</sub> | This study | N/A |
| Δ*dmdE*<sub>RI</sub>Δ*dddP*<sub>RI</sub>Δ*dddQ*<sub>RI</sub> | This study | N/A |
| **Recombinant DNA** | | |
| pHGM01 | Jin et al, 2013 | N/A |
| pHG101 | Wu et al, 2011 | N/A |
| pHG101-*dmdA* | This study | N/A |
| pHG101-*dmdR* | This study | N/A |
| pHG101-*dmdR*<sub>RI</sub> | This study | N/A |
| pHG101-*dmdE*<sub>RI</sub> | This study | N/A |
| pHG101-*dmdEF*<sub>RI</sub> | This study | N/A |
| pHGE-P<sub>tac</sub> | Luo et al, 2013 | N/A |
| pHGE-P<sub>tac</sub>-*acuI* | This study | N/A |
| pHGE-P<sub>tac</sub>-*dmdA*<sub>RI</sub> | This study | N/A |
| pHGE-P<sub>tac</sub>-*dmdEF*<sub>RI</sub> | This study | N/A |

| Reagent/resource | Reference or source | Identifier or catalog number |
|---|---|---|
| pHGE-P$_{tac}$-dddT | This study | N/A |
| pET28a(+) | Novagen | 69864-3 |
| pGEM-T | Promega | A1360 |
| **Antibodies** | | |
| Rabbit anti-DmdA | QWbio | N/A |
| Goat anti-rabbit HRP | Bioss | Bs-0295G-HRP |
| **Oligonucleotides and other sequence-based reagents** | | |
| PCR primers | This study | Appendix Table S6 |
| **Chemicals, enzymes and other reagents** | | |
| 2*Phanta Max Master Mix | Vazyme | P525 |
| Ready-to-use Seamless cloning Master Mix | BBI | B632219 |
| Dimethylsulfoniopropionate | TCI | C1240 |
| Acrylate | Sigma | 408220 |
| Adenosine triphosphate | Abmole | M9072 |
| 3,3'-Diaminobenzidine tetrahydrochloride | Yeasen | 36201ES03 |
| Coenzyme A Hydrate | Solarbio | C9640 |
| Ovalbumin | Cytiva | 28403841B |
| Conalbumin | Cytiva | 28403841 A |
| Catalase | Sigma | C30 |
| **Software** | | |
| MAGA 7 | www.megasoftware.net | |
| GraphPad Prism 9 | https://www.graphpad.com | |
| **Other** | | |
| HPLC 20 A | Shimadzu | |
| GC 2030 | Shimadzu | |
| Monolith NT.115 | NanoTemper | |

## Bacterial strains and growth conditions

Bacterial strains and plasmids used in this study are listed in Appendix Table S5. Routinely, *Escherichia coli* strains WM3064 and BL21 (DE3) were grown in Lysogeny broth (LB) medium at 37 °C. *R. pomeroyi* DSS-3 was grown in marine broth 2216E (Difco) medium at 30 °C for genetic manipulation. *R. lacuscaerulensis* ITI-1157 strains were grown at 40 °C for all experiments in the same medium as *R. pomeroyi* DSS-3 strains. Growth medium was supplemented with the following antibiotics as appropriate: gentamicin (Gm), 15 μg/ml; 2,6-diaminopimelic acid (DAP), 0.3 mM; kanamycin, 50 μg/ml, ampicillin, 50 μg/ml. For all other experiments, *R. pomeroyi* DSS-3 and *R. lacuscaerulensis* ITI-1157 strains were grown in minimal medium with NH$_4$Cl (1 mM) as nitrogen source and DMSP (6 mM for *R. pomeroyi* DSS-3 and 5 mM for *R. lacuscaerulensis* ITI-1157), acrylate (3 mM) or regular carbon mixture (CM, 5.6 mM glucose; 5.6 mM fructose; 8.5 mM succinate; 11.4 mM pyruvate; 10.9 mM glycerol; 17 mM acetate)

(Chen et al, 2011) as carbon source. Succinate (10 mM) and DMSP (6 mM) or acrylate (3 mM) were used as carbon sources in minimal medium to examine the toxicity of acrylate in *R. pomeroyi* DSS-3. Succinate (1 mM) and DMSP (5 mM) were used as carbon sources in minimal medium to demonstrate the accumulation of toxic catabolites in *R. lacuscaerulensis* ITI-1157 derived mutants.

## Mutagenesis and complementation

The *att*-based fusion PCR method previously described (Jin et al, 2013) was used to construct in-frame deletion strains of *R. pomeroyi* DSS-3 and *R. lacuscaerulensis* ITI-1157 with moderate modification with specific primers (Appendix Table S6). In brief, the fusion of two fragments flanking the target gene was introduced into *att*-based mutagenesis plasmid pHGM01 (Jin et al, 2013) by site-directed recombination, and transformed into *E. coli* WM3064. The resultant plasmid was transferred into recipient strains via conjugation. In-frame deleted mutants were screened as in (Jin et al, 2013), and verified by PCR sequencing. The Δ*ddd* mutant were generated by knocking out *dddW*, *dddP* and *dddQ* successively.

For complementation of gene mutants, fragments containing the target genes and their native promoters were amplified by PCR and cloned into pHG101 (Wu et al, 2011) with specific primers (Appendix Table S6). Alternatively, the target genes were amplified and cloned into pHGE-P$_{tac}$ (Luo et al, 2013) under the control of IPTG-inducible promoter P$_{tac}$. The resulting complementation vectors were transferred into the host bacteria by conjugation via *E. coli* WM3064 and verified by PCR and sequencing. Complementing vector of pHG101-*dmdR* was used to generate site-directed mutations of *dmdR*, using the PCR-based method and verified by sequencing. The mutant *dmdR* vectors were transferred into Δ*dmdR*, respectively, via conjugation. Primers used for complementation were list in Appendix Table S6.

## Quantification of DMS and MeSH by gas chromatography

Cells pre-grown to late-exponential phase in the marine broth 2216E medium were treated as in (Wang et al, 2023) and incubated in minimal medium (see above) with 6 mM DMSP as the sole carbon source in gas-tight vials sealed with polytetrafluoroethylene/silicone septa at 30°C for 42 h. The culture headspaces were assayed for DMS and MeSH production on a gas chromatography instrument (GC) (GC-2030, Shimadzu, Japan) following the method in (Curson et al, 2017; Liu et al, 2018). Cells were lysed by ultrasonication and the protein content was measured using the Pierce BCA Protein Assay Kit (Thermo, America). DMS and MeSH production is expressed as nmol min$^{-1}$ mg protein$^{-1}$.

## Real-time qPCR (RT-qPCR) analysis and Reverse Transcription PCR (RT-PCR)

Bacteria were precultured in 2216E medium at 30 °C (40 °C for *R. lacuscaerulensis* ITI-1157 strains) to OD$_{600}$ of 0.6. Cells for total RNA extraction were prepared as those for transcriptome sequencing in (Li et al, 2021) with different carbon sources (CM, DMSP or acrylate) induction for 4 h. Total RNA was extracted using RNeasy Mini Kit (Qiagen, Germany). RT-qPCR was performed as in (Li et al, 2021) with specific primers (Appendix

Table S6). For RT-qPCR of *dmdR*, qPCR primers were specifically designed to target the 5' region of *dmdR* (nucleotides +30 to +119 relative to the transcription start site, TSS). This region remains intact in the Δ*dmdR* strain, retaining a 120-bp segment downstream of the TSS (see Appendix Table S6 for primer details). For RT-qPCR in oxidative stress, wild-type *R. pomeroyi* DSS-3 and *R. lacuscaerulensis* ITI-1157 were exposed to 0.5 mM $H_2O_2$ and collected samples at different time points. For RT-PCR, the reverse-transcribed cDNA of *R. pomeroyi* DSS-3 or *R. lacuscaerulensis* ITI-1157 grown with 6 mM DMSP was used as PCR template, with specific primers (Appendix Table S6). The *R. pomeroyi* DSS-3 or *R. lacuscaerulensis* ITI-1157 genome was used as a control.

## Western blot assay

Cells were prepared exactly for RT-qPCR assays and sonicated. Protein concentrations of cell lyases were determined using Pierce BCA Protein Assay Kit. Equal amounts of protein (20.5 μg) were separated on SDS-PAGE gels and electrophoretically transferred to nitrocellulose membranes. The blotting membrane was incubated with anti-DmdA serum followed by a 1:5000 dilution of goat anti-rabbit immunoglobulin G-horse radish peroxidase for detection using ECL luminescence reagent (Absin). Images were visualized by autoradiography. The SDS-PAGE gel was stained by Coomassie brilliant blue to visualize the equal amount of loaded protein.

## Protein purification and site-directed mutation

*dmdR* was amplified from the *R. pomeroyi* DSS-3 genome and cloned into pET28a (+) vector with a C-terminal His-tag (Novagen, America). All site-directed mutations in DmdR to produce engineered variants for study were introduced using the PCR-based method and verified by DNA sequencing. Genes of *dmdR* homolog from *N. halophilus* LMG 25378 and *M. sedimentorum* OS208 were synthesized and cloned into pET28a (+) vector for purification. All of the DmdR proteins and variants were overexpressed in *E. coli* BL21(DE3) by adding 0.5 mM IPTG and purified using a $Ni^{2+}$-NTA column (GE Healthcare, America), then followed by gel filtration on a Superdex G200 column (30 cm × 10 mm; GE Healthcare, America). A 1 ml protein sample (1 mg/ml) was loaded onto the column and eluted with 10 mM Tris-HCl (pH 8.0), 100 mM NaCl at a flow rate of 0.4 ml/min. The molecular weight standards of 44 kDa (Ovalbumin) and 75 kDa (Conalbumin) (Cytiva, America) were used to determine the oligomerization state of DmdR and variants in solution.

## DNase I footprinting assays

The *dmdR* and *dmdA-acuI* intergenic region was cloned into plasmid pGEM-T (Promega, USA) and amplified using primers of T7 (FAM) and SP6 primers to generate a fluorescent FAM labeled probe. FAM-labeled probes were purified and quantified with Nano drop 2000C (Thermo, USA). DNase I footprinting assays were performed according to Wang et al (2012). For each assay, 400 ng probes were incubated with different amounts of DmdR in a total volume of 40 μl. Samples were successively extracted and precipitated with phenol/chloroform and ethanol, then dissolved in 30 μl MiliQ water. DNA ladder preparation, electrophoresis,

sequencing and data analysis were performed as described in (Wang et al, 2012), except that the GeneScan-LIZ600 size standard (Applied Biosystems) was used.

## Electrophoretic mobility shift assays (EMSAs)

EMSAs were performed in 20 μl reaction mixtures containing different concentrations of DmdR (10–800 nM) and selected DNA probes (7 nM, except for probe P2) in binding buffer (2 mM EDTA, 20 mM KCl, 0.5 mM DTT, 4% Ficoll-400, pH 8.0) as in (Zhang et al, 2017). Since probe P2 was too short to be visible at regular DNA probe concentration in EMSA, this was used at 237 nM. Probe DNA P1 (the 222 bp *dmdR-dmdA* intergenic space) was amplified from *R. pomeroyi* DSS-3 genomic DNA using specific primer set (Appendix Table S6). Probe DNA P2 (the 32 bp DmdR-protected region) were obtained by annealing the complementary oligonucleotides of the 32 bp DmdR-protected region (Appendix Table S6). The DNA fragment of the *dmdR-dmdA* intergenic region without *dmd box 1* or *dmd box* 2 or both was synthesized and used to amplify the probe P3, P4 and P5, respectively. For competing EMSA, PrpE enzymatic assay mixture was added to the DmdR (0.1 μM) and DNA probe P1 (4 nM) incubation mixtures at the specified concentration. The production of acryloyl-CoA with recombinant PrpE was determined by HPLC (Li et al, 2021; Wang et al, 2015). Incubation mixtures without recombinant PrpE (C-1) or with the K588A PrpE derivative (C-2) were used as controls, in which the absence of acryloyl-CoA was also determined by HPLC (Li et al, 2021; Wang et al, 2015). The intergenic regions between *dmdR* and *acuI* from *N. halophilus* LMG 25378 and *M. sedimentorum* OS208 were amplified using specific primer sets (Appendix Table S6). EMSAs between purified DmdR homologs and corresponding DNA probes (10 nM) were performed. For EMSAs with above probes, the reaction mixture was run on 12% non-denaturing polyacrylamide gels. The gels were pre-run to allow buffer equilibration before loading samples. On completion, the gel was stained by GelRed (0.005% v/v) in 1× TBE buffer at room temperature for 30 min, then visualized on a UV transilluminator.

## 5' rapid amplification of cDNA ends (RACE)

The *dmdA* and *dmdR* transcription start sites were identified by 5' RACE using a 5'-RACE kit (Sangon Biotech, China) according to the manufacturer's instructions. Total RNA was extracted from exponential phase *R. pomeroyi* DSS-3 cultures grown in minimal medium with 6 mM DMSP as sole carbon source. 0.5–2.0 μg of this RNA was used for cDNA synthesis using RT primers (Appendix Table S6). cDNA was processed and its 5'-end was amplified following the manufacturer's instructions using specific primers (Appendix Table S6). The final PCR product was cloned into the pMD18-T vector and products amplified for sequencing by PCR using M13 + /M13- primer sets. The transcription start site was determined as the first nucleotide following oligo-dC sequence.

## Microscale thermophoresis (MST)-binding assay

Purified DmdR or its site-directed mutants were labeled with the Large Volume Protein Labeling Kit RED-Tris-NTA 2nd Generation

(NanoTemper Technologies GmbH). Ligands were diluted in a range of concentrations and mixed with labeled DmdR or its site-directed variants at 25 °C in buffer containing 1×PBS (pH 7.4) and 0.05% Tween-20. Mixed samples were loaded into MonolithTM NT.115 Series capillaries (NanoTemper Technologies GmbH) and the thermophoresis was carried out on a Monolith NT.115 instrument (NanoTemper Technologies GmbH). Binding was measured with 60% LED power and "medium" MST power. $K_d$ values were obtained by fitting the MST data in the MO. Affinity Analysis software. The following ligands were used: DNA probe P2, 4.55 nM–149 μM; DNA probe P3, 0.37 nM–12 μM; DNA probe P4, 0.43 nM–14 μM; DMSP, 85 μM–2.8 M; acrylate, 92 μM-3 M; acryloyl-CoA, 122 nM–2 mM; propionyl-CoA, 46 nM–1.5 mM; 3-HP, 1.7 μM–55 mM. Competing MSTs with addition of 0.75 μM effector candidate (DMSP, acrylate, acryloyl-CoA, propionyl-CoA or 3-HP) to labeled DmdR (0.25 μM) were titrated against DNA probe P2 (4.57 nM–150 μM).

## Heterogeneous expression and DMSP concentration measurement

The $dmdE_{Rl}$-$dmdF_{Rl}$ coding region was amplified from *R. lacuscaerulensis* ITI-1157 using specific primer sets (Appendix Table S6) and cloned into pHGE-P$_{tac}$ vector driven by IPTG (Luo et al, 2013). These vectors generated were transformed individually into *E. coli* WM3064. The DMSP transporter gene $dddT$ was amplified from *Halomonas* strain and cloned into the same vector as a positive control. The intracellular DMSP levels were measured after culturing *R. lacuscaerulensis* ITI-1157 wild-type, $\Delta dmdE_{Rl}$, and its genetic complementation strains in minimal medium with 5 mM DMSP as sole carbon source for 96 h. For measuring intracellular DMSP in *E. coli* WM3064, cells harboring pHGE-P$_{tac}$-$dmdEF_{Rl}$, pHGE-P$_{tac}$-$dddT$, or empty vector and without vector were cultured in LB medium supplemented with 5 mM DMSP and 0.3 mM DAP. After IPTG induction for 10 h, cells were harvested to measure the intracellular DMSP concentrations. Complete chemical DMSP conversion to gaseous DMS was performed using 10 M NaOH and measured by GC as above.

## Peroxidase activity assays and ROS measurement

*R. lacuscaerulensis* ITI-1157 $dmdF$ was amplified and cloned into the pET28a (+) vector with a C-terminal His-tag (Novagen, America). 25 μg DmdF was loaded in non-denaturing polyacrylamide gels. The peroxidase activity of purified DmdF was detected as described previously (Wayne and Diaz, 1986). Catalase (~440 μg) was stained as a positive control. The consumption of $H_2O_2$ by purified DmdF is quantified using xylenol orange assay (Wolff, 1994). The liberation of oxygen was visualized by inoculating 20 μl DmdF (10 mg/ml) to 200 μl $H_2O_2$ (1.76 M). $H_2O_2$ without inoculation or with 20 μl protein buffer, in which DmdF is dissolved, were used as negative controls. The ROS levels of *R. lacuscaerulensis* ITI-1157 wild-type and $\Delta dmdF$ strains grown on DMSP were measured by commercial Reactive Oxygen Species Assay Kit (Yeasen, China) based on fluorescence dye 2,7-dichlorodi-hydrofluorescein diacetate following the manufacturer's instruction. Strains were inoculated into minimal medium with 5 mM DMSP. Equal cell volumes were harvested at indicated time points for probe treatment and fluorescence measurement.

## Bioinformatics

The $dmdR$-$dmdA$ intergenic regions of 32 Roseobacters were aligned using ClustalW embedded in MEGA version 7.0 and a weight matrix for the consensus sequences was generated using WebLogo (http://weblogo.berkeley.edu/). The maximum-likelihood phylogenetic tree based on protein sequences of DmdA homologs from 53 Roseobacters, whose genomes are available on NCBI, was constructed using MEGA version 7.0 and the tree topology was checked by 1000 bootstrap replicates. Homologs of $dmdR$, $dmdA$ and $acuI$ were retrieved from NCBI representative bacterial genomes using hmmer method. For $dmdR$, $dmdE$ and $dmdF$, hmm databases were used with a hmmsearch e-value of $1e^{-10}$ and only homologues with ≥30% amino acid identity and ≥70% coverage to the reference DmdR sequences were counted. For $acuI$, the hmm module was downloaded from KofamKOALA (https://www.genome.jp/tools-bin/kofamsearch, query ID: K19745) and was then used to search homologues with a hmmsearch e-value of $1e^{-10}$. For $dmdA$, we used a more stringent cutoff (hmmsearch e-value < $1e^{-30}$) as in our previous paper (Liu et al, 2022) and only homologues with ≥40% amino acid identity and ≥70% coverage to ratified sequences were counted. The three homologous genes with the distance less than 10 open reading frames were regarded as clustered. A maximum-likelihood phylogenetic tree based on DmdR homologs protein sequences was constructed using MEGA version 7.0 and the tree topology was checked by 1000 bootstrap replicates.

The relative abundance/expression of $dmdR$, $dmdE$, $dmdF$, $dmdA$ and $acuI$ genes were estimated by analysing the *Tara* Oceans datasets using the online webserver Ocean Gene Atlas (https://tara-oceans.mio.osupytheas.fr/ocean-gene-atlas/) (Vernette et al, 2022). Briefly, the hmm module of each gene was submitted to the webserver to detect homologues in OM-RGCv2 metagenomes/metatranscriptomes (Salazar et al, 2019) using the hmmer method. The relative abundance of each gene was normalized to the average relative abundance of 10 conserved single-copy marker genes as in Liu et al (2022) and was expressed as percent of prokaryotic cells. Ten marker genes were retrieved using the method detailed above and a hmmsearch e-value of <$1e^{-10}$. The relative expression of each gene was expressed as percent of mapped reads.

## Data availability

This study includes no data deposited in external repositories.

The source data of this paper are collected in the following database record: biostudies:S-SCDT-10_1038-S44318-026-00706-2.

## Peer review information

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

## Acknowledgements

This work was supported by the National Key R & D Program of China (2022YFC2807501, 2024YFC2816000), the National Science Foundation of China (W2441012, 32330001, 92251303, 42176156, 42276102), the Fundamental Research Funds for the Central Universities (202172002), Major Scientific and Technological Innovation Project (MSTIP) of Shandong Province (2019JZZY010817), the Taishan Scholars Program of Shandong Province, China (tspd20240806, tsqn202306092), the National Science Foundation of Shandong Province (ZR2025MS308, ZR2024YQ076), and the Biotechnology and Biological Sciences Research Council, UK (BB/X005968), Natural Environmental Research Council, UK, (NE/P012671, NE/S001352, NE/X000990 and NE/X014428 with DLS) and the Leverhulme trust (RPG-2020-413) grants.

## Author contributions

**Hui-Hui Fu**: Conceptualization; Formal analysis; Funding acquisition; Investigation; Writing—original draft; Project administration; Writing—review and editing. **Ming-Chen Wang**: Resources; Data curation; Software; Investigation; Methodology. **Zhi-Qing Wang**: Resources; Investigation; Methodology. **Yu-Han Sang**: Investigation; Methodology. **Zhen-Kun Li**: Investigation; Methodology. **Fei-Fei Li**: Investigation. **Jia-Rong Liu**: Resources; Investigation; Methodology. **Qi-Long Qin**: Data curation; Software. **Xiao-Yu Zhu**: Data curation; Software. **Na Wang**: Investigation; Methodology. **Jin-Jian Wan**: Resources; Investigation; Methodology. **Zhao-Jie Teng**: Data curation; Software. **Wei-Peng Zhang**: Data curation; Software. **Andrew J Gates**: Writing—review and editing. **Chun-Yang Li**: Funding acquisition; Writing—review and editing. **Jonathan D Todd**: Conceptualization; Funding acquisition; Project administration; Writing—review and editing. **Yu-Zhong Zhang**: Conceptualization; Funding acquisition; Project administration; Writing—review and editing.

Source data underlying figure panels in this paper may have individual authorship assigned. Where available, figure panel/source data authorship is listed in the following database record: biostudies:S-SCDT-10_1038-S44318-026-00706-2.

## Disclosure and competing interests statement

The authors declare no competing interests.

