## [Peer Review File · The EMBO Journal]

Regulation of DMSP organosulfur cycling in ubiquitous *Roseobacter* marine bacteria

Hui-Hui Fu, Ming-Chen Wang, Zhi-Qing Wang, Yu-Han Sang, Zhen-Kun Li, Fei-Fei Li, Jia-Rong Liu, Qi-Long Qin, Xiao-Yu Zhu, Na Wang, Jin-Jian Wan, Zhao-Jie Teng, Wei-Peng Zhang, Andrew Gates, Chun-Yang Li, Jonathan Todd, and Yu-Zhong Zhang

Corresponding authors: Yu-Zhong Zhang (zhangyz@sdu.edu.cn) , Jonathan Todd (jonathan.todd@uea.ac.uk)

Review Timeline:

Submission Date:	15th Aug 25
Editorial Decision:	7th Oct 25
Revision Received:	6th Nov 25
Editorial Decision:	10th Dec 25
Revision Received:	19th Dec 25
Accepted:	9th Jan 26

Editor: Yehu Moran

Transaction Report:

Dear Prof. Zhang,

Thank you for submitting your manuscript for consideration by the EMBO Journal. It has now been seen by two referees whose comments are shown below.

Given the referees' positive recommendations, I would like to invite you to submit a revised version of the manuscript, addressing the comments of both reviewers. I should add that it is EMBO Journal policy to allow only a single round of major revision, and acceptance of your manuscript will therefore depend on the completeness of your responses in this revised version.

Based on past experience, I would strongly recommend that after consulting with your co-authors you will send to me via email a revision plan within a few weeks. This could significantly help in making the revision process more efficient and to set realistic expectations between the authors and the editor.

Thank you for the opportunity to consider your work for publication. I look forward to your revision.

Yours sincerely,

Yehu Moran
Academic Editor
The EMBO Journal

We realize that it is difficult to revise to a specific deadline. In the interest of protecting the conceptual advance provided by the work, we recommend a revision within 3 months (5th Jan 2026). Please discuss the revision progress ahead of this time with the editor if you require more time to complete the revisions.

Referee #1:

Hui-Hui Fu, Ming-Chen Wang, and colleagues investigate how marine Roseobacter group bacteria regulate dimethylsulfoniopropionate (DMSP) catabolism, a process central to global sulfur cycling and the formation of the climate-active gas dimethylsulfide (DMS). They identify DmdR as a key regulator that senses intracellular DMSP levels and acts as a molecular switch, shifting between DMSP cleavage alone and a combined cleavage-demethylation response under high intracellular DMSP exposure. The regulator also coordinates pathways for detoxification of acryloyl-CoA, a toxic intermediate in the cleavage pathway, as well as hydrogen peroxide (H₂O₂), a byproduct of the demethylation pathway. This regulatory system, widespread in marine environments, highlights an important mechanism that links microbial metabolism to ocean biogeochemistry and climate regulation.

The paper is interesting and effectively combines basic biochemistry with molecular biology to unravel several components of this complex pathway. The authors convincingly identify the promoter regions targeted by DmdR and provide supporting evidence for their conclusions.

Overall, I find this research valuable and suitable for publication, with a few minor comments that could further improve the clarity of the manuscript:

- The abstract was somewhat challenging to follow and required multiple readings to fully grasp the main points. I understand that the work presents a rich set of results that are difficult to condense into such a short format, but improving the clarity of the abstract would significantly enhance its accessibility and impact.

- Figure 5 is very helpful in illustrating the complex regulatory network mediated by DMSP levels and DmdR. Improving the figure's quality and presentation would make it even clearer and strengthen the paper overall.

- Lines 436-437: "DMSP (6 mM/5 mM)" - this needs small clarification. Should it be 5 mM or 6 mM, a range (5-6 mM), or a ratio (6/5 mM)?

Referee #2:

The manuscript by Fu & Wang et al. identifies the regulator DmdR, which depending on intracellular concentrations of dimethylsulfoniopropionate (DMSP), controls the switch between DMSP demethylation and cleavage pathways in the model Roseobacter bacterium *Ruegeria pomeroyi*. The manuscript presents novel findings supported by an impressive dataset generated using a combination of approaches, from bacterial genetics to biochemical assays. DMSP is abundant in marine environments and serves as an important source of reduced carbon and sulfur for marine bacteria, through the demethylation and cleavage pathways, respectively. The factors regulating the switch between these pathways are not fully described. Therefore, the findings of this study are novel and relevant.

General comments:

1. In general, some ideas could be expanded to improve clarity and presentation of the results. Given the large volume of data, smoother transitions between subheadings would help guide the reader. The discussion could also be strengthened by integrating additional relevant literature and by providing a broader perspective on the implications of this regulatory 'switch' for the sulfur cycle.
2. The introduction of the term "MRB" seems redundant with existing terminology in the field. Members of the Roseobacteraceae family that inhabit marine environments are already referred to as Roseobacters or the Roseobacter clade (e.g., Brinkhoff, T.,

Giebel, H.A., & Simon, M. 2008. Diversity, ecology, and genomics of the Roseobacter clade: a short overview. *Archives of Microbiology*, 189(6):531-539). If the authors wish to use the term MRB, they should explain the rationale for coining a new term when "Roseobacters" already denotes marine representatives of this family.

3. I strongly recommend revising the manuscript for sentence clarity and conciseness. Many sentences combine several ideas, which makes them difficult to follow. For instance: "Bacterioplankton, particularly MRB, can import and concentrate dissolved DMSP to high millimolar levels for its antistress properties and/or drive two catabolic pathways..."

This sentence is long and complex, consider splitting it to convey one or two ideas per sentence. Similarly: "By contrast, both Δ acul and Δ dmdA Δ acul strains could not grow on acrylate, and phenotypes were rescued by expression of acul (but with an extended 89 h lag phase in Δ dmdA Δ acul) but not dmdA in trans (Fig. 1g and 1h)."

This contains multiple observations that would benefit from being divided into two sentences. Such issues recur throughout the manuscript and make it challenging to read.

4. Please also revise the repeated use of "and/or" throughout the text. Indicate whether there is a biological distinction between "and" and "or" cases. For example, is it significant if Roseobacters possess both dmdA and ddd genes, or if they have only one? Clarifying these distinctions will greatly improve readability and precision.

5. Finally, the manuscript would benefit from language editing for grammar and phrasing. For example:

- "Growth of Δ dmdA and Δ acul strains on DMSP as a sole carbon source."
- "We propose **that** DmdR represses dmdA-acul transcription."

Specific comments:

-Line 44: The meaning of "(micro)organisms" is unclear. If this distinction is important, please clarify the respective contributions of microbes and metazoans.

-Lines 49-51, 61-62: Additionally, DMSP demethylation via dmdA can serve as a methyl group donor to fuel the bacterial methionine cycle, and micromolar concentrations have been reported to influence bacterial physiological functions (Sperfeld & Narváez-Barragán, et al., 2024).

-Lines 82-90: The authors describe the single mutants as showing 'slightly impaired growth', but there is a reduction in the maximum yield reached. In addition, the double mutant Δ dmdA Δ acul appears to exhibit a small OD increase at early timepoints, suggesting that growth is impaired but not completely abolished. Quantifying growth parameters (e.g., growth rate, doubling time, lag phase duration) would help make these differences more precise and clearer.

-Line 98-99: This appears to be a very strong claim, given that the Δ dmdA mutant is still able to grow on DMSP as a carbon source (Fig. 1C). Moreover, in other Roseobacters, DMSP-demethylation enzymes independent of DmdA have been reported, although are active only under stress conditions (Narváez-Barragán et al., 2025). This statement should therefore be toned down.

-Lines 102-110: In Supplementary Fig. 1C, dmdA appears to be more highly expressed in the dmdR mutant even without DMSP addition. Is the difference between CM and 30 μ M DMSP statistically significant?

-Lines 136-137 and 143: The band shifts in the EMSAs are difficult to distinguish in the figures. Would it be possible to improve their visualization by including a gel with better separation, enlarging the relevant region, or adding markers/lines to highlight the changes?

-Lines 151-152: According to Fig. 3B, E and F, P1 and P4 show a similar trend, in contrast with P3. Does this suggest that box 1 plays a more central role in the interaction?

-Lines 162-170: The expression of dmdA appears to be higher when bacteria are exposed to acrylate (Fig. 2C). Is there an induction of dmdA or of genes associated with candidate dmd boxes under acrylate exposure in the Δ ddd mutant?

-Lines 208-210: Are the differences between the site-directed mutations statistically significant? Do the data indicate that certain residues are more critical for the interaction than others?

-Lines 238-242: Because high intracellular concentrations of DMSP are required, could this molecule also function as an osmoprotectant? Moreover, micromolar concentrations of extracellular DMSP have been reported to activate the demethylation pathway in Roseobacters (Wirth et al., 2020, Sperfeld & Narváez-Barragán, et al., 2024).

-Lines 318-320: Are there examples where DMSP catabolites can be toxic? In addition, could the growth differences observed between Δ dmdE and Δ dmdF mutants and the WT on Succinate + DMSP reflect the ability of the WT to utilize both substrates as carbon sources, while the mutants cannot utilize DMSP (Fig. 6A)?

-Lines 342-345: These are very interesting results, the discussion would benefit from including literature on DMSP demethylation and oxidative stress in bacteria. For example, in response to oxidative stress, *Ruegeria* DMSP metabolism has been reported to shift from the demethylation to the cleavage pathway, and KatG expression is influenced by DMSP exposure (Wang et al., 2022). In environmental datasets, dmdA and katG are reported to be co-expressed (Varaljay et al., 2015). Moreover, DMSP demethylases have been shown to be induced by oxidative stress in *P. inhibens*, which seems to be conserved in other bacteria (Narváez-Barragán et al., 2025). Since oxidative stress is also related to the switch between demethylation and cleavage pathways in *R. pomeroyi*, it would be helpful to test whether dmdR is upregulated by H₂O₂.

-Line 352: Could the higher abundance of dmdR/dmdF/dmdE transcripts in the mesopelagic zone be related to oxidative stress or substrate availability at depth?

-A brief discussion on the implications of this regulatory switch toward the cleavage pathway for sulfur fluxes would strengthen the broader context of the study.

References:

Sperfeld, M., Narváez-Barragán, D.A., Malitsky, S., Frydman, V., Yuda, L., Rocha, J., and Segev, E. (2024). Algal methylated compounds shorten the lag phase of *Phaeobacter inhibens* bacteria. *Nat Microbiol*, 1-16. <https://doi.org/10.1038/s41564-024->

01742-6.

Narváez-Barragán, D.A., Sperfeld, M., and Segev, E. DmdA-independent lag phase shortening in *Phaeobacter inhibens* bacteria under stress conditions. *The FEBS Journal*. doi:10.1111/febs.70128.

Wirth JS, Wang T, Huang Q, White RH & Whitman WB (2020) Dimethylsulfoniopropionate sulfur and methyl carbon assimilation in *Ruegeria* species. *MBio* 11, e00329-20.

Wang, T., Huang, Q., Burns, A. S., Moran, M. A., & Whitman, W. B. (2022). Oxidative Stress Regulates a Pivotal Metabolic Switch in Dimethylsulfoniopropionate Degradation by the Marine Bacterium *Ruegeria pomeroyi*. *Microbiology spectrum*, 10(6), e0319122. <https://doi.org/10.1128/spectrum.03191-22>

Varaljay, V.A., Robidart, J., Preston, C.M., Gifford, S.M., Durham, B.P., Burns, A.S., Ryan, J.P., Marin III, R., Kiene, R.P., Zehr, J.P., et al. (2015). Single-taxon field measurements of bacterial gene regulation controlling DMSP fate. *ISME J* 9, 1677- 1686. doi:10.1038/ismej.2015.23.

Referee #1:

Thank you for your insightful and constructive comments and advice. We have now carefully addressed these comments and made appropriate revisions to the manuscript that we feel have greatly improved its quality and accessibility. Please find our responses to your comments.

1. The abstract was somewhat challenging to follow and required multiple readings to fully grasp the main points. I understand that the work presents a rich set of results that are difficult to condense into such a short format, but improving the clarity of the abstract would significantly enhance its accessibility and impact.

Response: Thank you for this comment which we completely agree with. We have slightly extended the length of our abstract but feel that it is far clearer for the reader. Please see below:

“Dimethylsulfoniopropionate (DMSP) catabolism by marine **Roseobacters** is important for **global biogeochemical** cycling and **the** climate. Many **Roseobacters** contain **competing** DMSP demethylation and cleavage pathways, **but** only cleavage produces the climate-cooling gas dimethylsulfide. We identify the “switch” regulator in **Roseobacters**, DmdR, that represses **the transcription of** DMSP demethylation (*dmdA*, encoding DMSP demethylase), cleavage (*acul*, encoding acryloyl-CoA reductase) and often novel oxidative stress protection (*dmdEF*, *dinB*) genes under low intracellular DMSP levels. **Increased DMSP levels induce DMSP cleavage and accumulation of cytotoxic acryloyl-CoA.** DmdR binds acryloyl-CoA as its effector and derepresses *dmdA-acul* transcription to stimulate acryloyl-CoA **catabolism and DMSP demethylation.** **Co-upregulation of the novel peroxidase DmdF and likely DmdE and DinB counteract oxidative stress associated with DMSP demethylation.** Thus, DmdR, with DmdR-independent regulation of DMSP cleavage, likely balances cellular DMSP levels to allow its antistress functions, but accelerated demethylation and catabolism of toxic intermediates at higher DMSP levels. **In abundant marine bacteria lacking *dmdA*, DmdR still likely controls acryloyl-CoA catabolism/detoxification.** DmdR and **DmdEF** are widespread in Earth’s oceans and important in biogeochemical cycling and climate-active gas production.”

2. Figure 5 is very helpful in illustrating the complex regulatory network mediated by DMSP levels and DmdR. Improving the figure's quality and presentation would make it even clearer and strengthen the paper overall.

Response: Thank your positive feedback on Fig. 5 and for the constructive suggestion. In response, we have further improved the figure by illustrating the regulatory mechanism - specifically, how DmdR binds to DNA and how this interaction is attenuated by the effector acryloyl-CoA. Additionally, the overall layout has been reformatted to enhance clarity. The revised version of Fig. 5 is provided below as it is in the revised manuscript.

Fig. 5 DmdR mediated regulation of DMSP catabolism. a, When DMSP concentration is low, the DmdR dimer binds to *dmd box* sequences in the promoter region of *dmdA-*acul** (including *dmdEF* in many cases), resulting in repression of DMSP demethylation, oxidative stress protection and downstream DMSP cleavage (catalyzed by enzyme *AcuI*), ensuring conditions that favor its intracellular accumulation. **b,** When DMSP concentration increases, *ddd* gene expression is increased, leading to enhanced DMSP cleavage and accumulation of the DmdR effector acryloyl-CoA through the action of *PrpE*. DmdR binds to its acryloyl-CoA effector. Holo-DmdR with lower affinity to *dmd boxes* dissociates from the *dmdA-*acul** promoter derepressing transcription. This upregulates expression of the demethylation pathway and the *AcuI* enzyme of the DMSP cleavage pathway, that detoxifies acryloyl-CoA and enables DMSP's use as a carbon and sulfur source. Enhanced DMSP demethylation results in reactive oxygen species (ROS) production and oxidative stress, which can be alleviated via the action of the peroxidase *DmdF* and, potentially, *DmdE* and *DinB*, whose expression is also derepressed.

3. Lines 436-437: "DMSP (6 mM/5 mM)" - this needs small clarification. Should it be 5 mM or 6 mM, a range (5-6 mM), or a ratio (6/5 mM)?

Response: Thank you for pointing this out. The DMSP concentration used for the growth of *R. pomeroyi* DSS-3 is 6 mM, and for *R. lacuscaerulensis* ITI-1157 it is 5 mM. We have changed the description on lines 454-457 as follows:

"For all other experiments, *R. pomeroyi* DSS-3 and *R. lacuscaerulensis* ITI-1157 strains were grown in minimal medium with NH_4Cl (1 mM) as nitrogen source and

DMSP (6 mM for *R. pomeroyi* DSS-3 and 5 mM for *R. lacuscaerulensis* ITI-1157), acrylate (3 mM) or regular carbon mixture...”

Referee #2:

We sincerely appreciate the time and effort this reviewer put into our manuscript. The insightful and constructive feedback has enabled us to produce a much-improved manuscript. Please find our responses to your comments.

General comments:

1. In general, some ideas could be expanded to improve clarity and presentation of the results. Given the large volume of data, smoother transitions between subheadings would help guide the reader. The discussion could also be strengthened by integrating additional relevant literature and by providing a broader perspective on the implications of this regulatory 'switch' for the sulfur cycle.

Response: Thank you for the suggestions to improve our manuscript. We have carefully been over the manuscript to improve its clarity, make smoother transitions between sections, and to strengthen the discussion via addition of relevant references and by summarizing the implications stemming from the study. The specific details of our changes are included in the answering of the questions below.

2. The introduction of the term "MRB" seems redundant with existing terminology in the field. Members of the Roseobacteraceae family that inhabit marine environments are already referred to as Roseobacters or the Roseobacter clade (e.g., Brinkhoff, T., Giebel, H.A., & Simon, M. 2008. Diversity, ecology, and genomics of the Roseobacter clade: a short overview. Archives of Microbiology, 189(6):531-539). If the authors wish to use the term MRB, they should explain the rationale for coining a new term when "Roseobacters" already denotes marine representatives of this family.

Response: Thank you for your comment. We agree that "Roseobacters" is the established and widely-recognized term in the field. As suggested, we have replaced the term "MRB" with "Roseobacters" throughout the revised manuscript.

3. I strongly recommend revising the manuscript for sentence clarity and conciseness. Many sentences combine several ideas, which makes them difficult to follow. For instance: "Bacterioplankton, particularly MRB, can import and concentrate dissolved DMSP to high millimolar levels for its antistress properties and/or drive two catabolic pathways..." This sentence is long and complex, consider splitting it to convey one or two ideas per sentence. Similarly: "By contrast, both Δ acuI and Δ dmdA Δ acuI strains could not grow on acrylate, and phenotypes were rescued by expression of acuI (but with an extended 89 h lag phase in Δ dmdA Δ acuI) but not dmdA in trans (Fig. 1g and 1h)." This contains multiple observations that would benefit from being divided into two sentences. Such issues recur throughout the manuscript and make it challenging to read.

Response: We again thank this reviewer for their suggestions to make our manuscript easier to read. Both of the highlighted sentences have been split into two, as detailed

below and we have carefully been over the manuscript to seek out and improve other such examples.

-Line 47-54: “Bacterioplankton, particularly **Roseobacters**, can import and concentrate dissolved DMSP to high millimolar levels¹³ allowing it to act as an antistress compound **in e.g. osmoprotection**^{10,14}. Many such bacteria can also catabolise this DMSP via two possible pathways (Fig. 1a)^{8,9,15}. Bacterial DMSP demethylation is initiated by DmdA and can be used for carbon and sulfur (via methanethiol, MeSH) assimilation, **and serve as a methyl donor to fuel the methionine cycle**^{11,13,15,16}. **Note, DMSP demethylation has also been reported to** cause oxidative stress¹⁷⁻¹⁹.”

-Line 91-94: “By contrast, both $\Delta acuI$ and $\Delta dmdA\Delta acuI$ strains could not grow on acrylate. **Growth on acrylate was** rescued by **the** expression of **cloned** *acuI* (but with an extended lag phase in $\Delta dmdA\Delta acuI$) but not *dmdA* (Fig. 1g and 1h).”

4. Please also revise the repeated use of "and/or" throughout the text. Indicate whether there is a biological distinction between "and" and "or" cases. For example, is it significant if Roseobacters possess both *dmdA* and *ddd* genes, or if they have only one? Clarifying these distinctions will greatly improve readability and precision

Response: We have carefully been over the manuscript and removed every instance of “and/or” and thus have explained the implications.

5. Finally, the manuscript would benefit from language editing for grammar and phrasing. For example: • "Growth of $\Delta dmdA$ and $\Delta acuI$ strains on DMSP as a sole carbon source." • "We propose **that** DmdR represses *dmdA-acuI* transcription."

Response: We thank the reviewer for pointing out these issues. We have addressed all the issues raised and have thoroughly checked the manuscript to correct any grammatical and phrasing mistakes.

Specific comments:

1. Line 44: The meaning of "(micro)organisms" is unclear. If this distinction is important, please clarify the respective contributions of microbes and metazoans.

Response: Thank you for your comment. Since macroalgae, corals, and some plants produce significant DMSP levels we thought it more appropriate to consider both microbes and larger organisms. However, we agree that our use of the word “(micro)organisms” was unclear. Thus, we changed our description to “organisms” on lines 45-46 in the revised manuscript as follows:

“Marine **organisms** produce >8 billion tons of dimethylsulfoniopropionate (DMSP) annually¹⁻³,”

2. Lines 49-51, 61-62: Additionally, DMSP demethylation via *dmdA* can serve as a methyl group donor to fuel the bacterial methionine cycle, and micromolar concentrations have been reported to influence bacterial physiological functions

(Sperfeld & Narváez-Barragán, et al., 2024).

Response: Thank you for providing this important reference that we missed. We have added the methyl group donor role of DMSP demethylation to the revised manuscript on lines 50-53 and added this reference.

“Bacterial DMSP demethylation is initiated by DmdA and can be used for carbon and sulfur (via methanethiol, MeSH) assimilation, and serve as a methyl donor to fuel the methionine cycle^{11,13,15,16}.”

We also thank the reviewer for highlighting that micromolar DMSP concentrations shorten the log phase of the Roseobacter *Phaeobacter inhibens* (also reported in Sperfeld & Narváez-Barragán, et al., 2024). However, we feel that it will be confusing to the readers to integrate as suggested because on lines 64-67 of our manuscript here we write, “DMSP catabolic enzymes have K_m values in the millimolar range, consistent with a kinetic strategy to maintain high intracellular DMSP levels needed for its physiological function”. We believe that integrating the results of Sperfeld & Narváez-Barragán, et al. may cause confusion regarding the distinct themes of enzyme kinetics and physiological function. Therefore, we have decided to retain the original description to ensure clarity. Nevertheless, we have fully integrated the micromolar induction phenomenon into our manuscript, as detailed in our answer to question 10 from this reviewer, please see below.

3. Lines 82-90: The authors describe the single mutants as showing 'slightly impaired growth', but there is a reduction in the maximum yield reached. In addition, the double mutant $\Delta dmdA\Delta acul$ appears to exhibit a small OD increase at early timepoints, suggesting that growth is impaired but not completely abolished. Quantifying growth parameters (e.g., growth rate, doubling time, lag phase duration) would help make these differences more precise and clearer.

Response: Thank you for your suggestion. We have quantified the growth of corresponding strains and included the data in the revised manuscript on lines 85-88 as follows,

“Growth of the $\Delta dmdA$ and $\Delta acul$ strains on DMSP as a sole carbon source was slightly impaired with a reduced maximum yield compared to the wild type strain (OD_{max} of WT ≈ 0.20 , OD_{max} of $\Delta dmdA \approx 0.16$, OD_{max} of $\Delta acul \approx 0.16$, Fig. 1c and 1d), but growth was abolished in the $\Delta dmdA\Delta acul$ strain”

For the next comment, we believe that the small observed increase in OD for the $\Delta dmdA\Delta acul$ double mutant (and indeed for all strains) was likely due to residual intracellular nutrients from pre-growth in 2216E medium. In addition, no MeSH was detected with the double mutant $\Delta dmdA\Delta acul$ when grown on DMSP (see figure below), indicating the complete loss of demethylation activity. Furthermore, the absence of *acul* completely abolished growth on acrylate (Fig. 1g and h), indicating blockage of the DMSP cleavage pathway. Thus, we have not modified the text. If this reviewer feels modification is necessary of the text we will of course do so.

Fig. 1 DMSP catabolic pathways and the importance of *dmdA* and *acuI* in *R. pomeroyi* DSS-3.

a, The DMSP demethylation (in blue) and cleavage pathways (in salmon). The key enzymes in *R. pomeroyi* DSS-3 are indicated in bold. DMSP, dimethylsulfoniopropionate; DMS, dimethyl sulfide; MMPA, methylmercaptopropionate; THF, tetrahydrofolate; 3-HP, 3-hydroxypropionate. **b**, Organization of the *R. pomeroyi* DSS-3 *dmdA-acuI* operon and *dmdR*. **c-h**, Growth of wild-type *R. pomeroyi* DSS-3, $\Delta dmdA$, $\Delta acuI$, $\Delta dmdA\Delta acuI$ and genetically complemented strains in minimal medium with DMSP (6 mM, **c-e**) or acrylate (3 mM, **f-h**) as sole carbon source, respectively. The error bar represents standard deviation of triplicate experiments.

4. Line 98-99: This appears to be a very strong claim, given that the $\Delta dmdA$ mutant is still able to grow on DMSP as a carbon source (Fig. 1C). Moreover, in other Roseobacters, DMSP-demethylation enzymes independent of DmdA have been reported, although are active only under stress conditions (Narváez-Barragán et al., 2025). This statement should therefore be toned down.

Response: We sincerely thank the reviewer for this insightful comment and for bringing our attention to the recent study by Narváez-Barragán et al., 2025, which reports DmdA-independent DMSP demethylation in the Roseobacter *Phaeobacter inhibens*. In our study on *R. pomeroyi* DSS-3, the observed growth of the *dmdA* mutant on DMSP as a carbon source was supported by the cleavage pathway, rather than alternative demethylation routes since no growth is observed in the $\Delta dmdA\Delta acuI$ (see above). This was further confirmed by gas chromatography analysis, which detected no MeSH produced in the *dmdA* mutant (figure below), indicating that DmdA is the only DMSP demethylase in this strain. It is of course possible that there are other proteins with DMSP demethylation activity, but these were not called in to action under the conditions tested here. Thus, we prefer not to introduce factors to further complicate our story.

Figure for reviewers removed.

However, in light of the reviewer's valuable input, we do agree that the statement "there were no other pathways for DMSP-dependent carbon assimilation in *R.*

pomeroyi DSS-3” should be more carefully qualified. We have revised the text on lines 100-103 as follows and incorporated the reference of Narváez-Barragán et al., 2025 to acknowledge the presence of DmdA-independent DMSP catabolism in other Roseobacters.

“Nevertheless, these data are consistent with previous studies^{10,15,32} and confirmed that in *R. pomeroyi* DSS-3, unlike in some other Roseobacters⁴¹, *dmdA* and *acul* are essential for DMSP-dependent carbon assimilation, via demethylation and cleavage, respectively.”

5. Lines 102-110: In Supplementary Fig. 1C, *dmdA* appears to be more highly expressed in the *dmdR* mutant even without DMSP addition. Is the difference between CM and 30 μ M DMSP statistically significant?

Response: We assume this comment refers to Fig. 2C, which presents the relative expression of *dmdA* with or without the addition of DMSP or acrylate. The *dmdA* transcript level was indeed significantly elevated in the Δ *dmdR* mutant without DMSP addition, which is consistent with the role of DmdR as a repressor of *dmdA* transcription.

The difference between CM and 30 μ M DMSP is indeed statistically significant. Accordingly, we have now modified our statement to include the significance values (lines 130-134): “Note, there was still marginal but significant ($p = 0.018$ and 0.006 for *dmdA*; $p = 0.017$ for *acul* with increased acrylate levels; $p = 0.012$ and 0.0001 for *dmdR*) upregulation of *dmdA-acul* and *dmdR* transcription in Δ *dmdR* with increased DMSP/acrylate levels, implying that there may be other regulators/factors influencing their transcription.”

6. Lines 136-137 and 143: The band shifts in the EMSAs are difficult to distinguish in the figures. Would it be possible to improve their visualization by including a gel with better separation, enlarging the relevant region, or adding markers/lines to highlight the changes?

Response: We appreciate the reviewer’s comment regarding the EMSA band shifts. Upon re-examination, we believe the separation between the shift band and the free probe is clear. The shifted band migrates at a much higher level and is well-resolved from the free probe front. The DNA marker (Lane M) also provides a clear frame of reference for this separation. To directly address this point, we have indicated the shifted band with a black triangle in the revised Fig. 3, as shown below. We feel that this improves the clarity of the figure.

Fig. 3 DmdR binds to *dmd box* sequences in the *dmdR* and *dmdA-acuI* promoters. **a.** Schematic of DNA probes used in EMSAs. P1: the full-length *dmdR* and *dmdA-acuI* intergenic region; P2: the 32 bp DmdR-protected region revealed by DNase I footprinting (see Appendix Fig. S3b); P3: variant of P1 without *dmd box 1*; P4: variant of P1 without *dmd box 2*; P5: variant of P1 without the 32 bp DmdR-protected region, *i.e.* without both *dmd boxes*. **b.** EMSAs of DmdR titrated against P1 (7 nM). Lane 1, P1; lane 2 to 11 show addition of increasing quantities of DmdR (10, 20, 40, 60, 80, 100, 200, 400, 600, and 800 nM respectively). M: DNA marker. **c.** The partial intergenic space sequence containing the 32 bp DmdR-protected region (in bold). The *dmdA-acuI* and *dmdR* transcription start sites (TSSs) are in bold, italics and indicated with an arrow. -10 and -35 boxes are framed. The DNA sequences of *dmd box 1* and *dmd box 2* were marked in blue and red, respectively. **d-f,** EMSAs of DmdR titrated against P2 (237 nM), biotin-5'-labeled P3 (7 nM), and biotin-5'-labeled P4 (7 nM) as indicated. Lane 1, corresponding DNA probe; Lane 2 to 11 show addition of increasing quantities of DmdR (10, 20, 40, 60, 80, 100, 200, 400, 600, and 800 nM respectively). The shift band is indicated by black triangle. M: DNA marker. Data presented are from at least three independent experiments.

7. Lines 151-152: According to Fig. 3B, E and F, P1 and P4 show a similar trend, in contrast with P3. Does this suggest that box 1 plays a more central role in the interaction?

Response: We sincerely thank the reviewer for this insightful observation. The reviewer is correct in noting that P4 exhibited stronger binding to DmdR than P3 in our EMSA results. This is further supported by the MST data, which showed a slightly lower K_d value for DmdR binding to P4 compared to P3 (Appendix Fig. S3e and f), consistent with a potentially higher affinity for the promoter containing *dmd box 1*. However, as both EMSA and MST are *in vitro* binding assays, and we were unable to directly test the individual functional importance of the two *dmd boxes in vivo* due to their overlapping essential promoter elements - specifically, the -35 region of *dmdA* (*dmd box 1*) and the -10 regions of *dmdA* and *dmdR*, and the *dmdR* transcription start site (*dmd box 2*). We have chosen to refrain from making a

definitive conclusion regarding their relative importance in the physiological context. Therefore, as detailed in the manuscript on lines 157-163, we have conservatively stated that DmdR can bind either *dmd box*, without assigning a hierarchical role to either motif.

“The respective importance of these *dmd boxes* was not examined further *in vivo* because they overlapped with the -35 region of *dmdA* (*dmd box 1*) or the -10 regions of *dmdA* and *dmdR*, and the *dmdR* transcription start site (*dmd box 2*) predicted from 5’ rapid amplification of cDNA ends (RACE) analysis (Fig. 3c). These results are consistent with DmdR acting to repress transcription of *dmdA-acuI* and itself by binding to either *dmd box* in the promoter region to prevent transcriptional initiation.”

8. Lines 162-170: The expression of *dmdA* appears to be higher when bacteria are exposed to acrylate (Fig. 2C). Is there an induction of *dmdA* or of genes associated with candidate *dmd boxes* under acrylate exposure in the Δddd mutant?

Response: Thank you for your comment. You are right to note the higher *dmdA* expression under acrylate exposure. This is expected because acryloyl-CoA, the direct product of acrylate, is the authentic inducer of *dmdA-acuI* operon. Regarding the Δddd mutant, the key point is that this mutant blocks the conversion of DMSP to acrylate. This in turn prevents the endogenous production of the inducer from DMSP, which is why we see barely any induction of *dmdA* in Δddd mutant when DMSP is provided (Fig. 4a, included below). This also explains why DMSP, which generates acryloyl-CoA less efficiently due to it being catabolized via both demethylation and cleavage pathways, shows a weaker inductive effect compared to direct acrylate addition. i.e. there will be less acryloyl-CoA made from DMSP because a significant proportion of substrate will be channeled through the demethylation pathway. We appreciate your question and hope this clarifies our findings.

Fig. 4 Acryloyl-CoA is the DmdR effector. **a,** Relative transcript levels of *dmdA* in wild type DSS-3 and Δddd strains grown with CM or DMSP (6 mM). **b,** Relative transcript levels of *dmdA* in wild type DSS-3 and $\Delta acul$ strains grown with CM or acrylate (3 mM). **c,** MST analysis of DmdR binding to acryloyl-CoA. The acryloyl-CoA was titrated from 1.22 μ M to 2 mM. DmdR had a K_d of 9.61 ± 4.22 μ M for acryloyl-CoA. **d.** MST analysis of DmdR binding to DNA probe P2 in the presence of acryloyl-CoA (1.25 μ M). The DNA probe P2 was titrated from 381 nM to 12.5 mM. The K_d of DmdR for P2 dramatically increased in the presence of acryloyl-CoA to 2.30 ± 1.43 mM. **e.** Competing EMSA analysis of interactions between DmdR and DNA probe P1 (0.3 nM) in the presence of increasing amount of enzymatic assay mixture (EAM) of PrpE. C-1 and C-2, incubation mixture without recombinant PrpE and with K588A variant of PrpE, respectively. The shift band is indicated by black triangle. M: DNA marker. The error bar represents standard deviation of triplicate experiments. A two-sided Student's *t*-test was used to assess statistically significant differences (**, $p < 0.01$). Data presented are at least three independent experiments.

Furthermore, to directly address your concern, we measured the induction of *dmdA* using RT-qPCR. As shown in the figure below, the induction of *dmdA* by acrylate in the Δddd mutant was comparable to that in the wild type. This result is consistent with the role of Ddd enzymes in the cleavage pathway, indicating that their gene mutation does not affect acrylate catabolism.

Figure for reviewers removed.

9. Lines 208-210: Are the differences between the site-directed mutations statistically significant? Do the data indicate that certain residues are more critical for the interaction than others?

Response: We sincerely appreciate the reviewer for raising this question. Our analysis confirms statistically significant differences among the mutants. The D163A, S171A, and E179A mutations resulted in significantly lower *dmdA* transcript levels compared to the Q85A and E136A mutations. This indeed indicates that residues D163, S171, and E179 are likely more critical for the DmdR-effector interaction. We have now added the statistical analysis to Appendix Fig. S6d and stated this conclusion in the revised manuscript on lines 215-218 as follows:

“Of these site-directed mutations, D163A, S171A, and E179A exhibited lower *dmdA* transcript levels compared to Q85A and E136A, implying that D163, S171, and E179

were likely more critical for the interaction between DmdR and its effector.”

Supplementary Fig. 6 Identification of the acryloyl-CoA binding sites in the E-O domain of *R. pomeroyi* str. DSS-3 DmdR. **a**, The DmdR monomer has two domains: an N-terminal DNA binding domain (green) and C-terminal effector-binding and oligomerization domain (E-O domain, yellow). The DNA binding domain contains a winged helix-turn-helix motif (green). The C-terminal domain is arranged into an all- α -helical bundle (yellow). **b**, Surface representation of the domain-swapped dimeric structure of DmdR. DmdR forms a domain-swapped dimer through the E-O domain. White dotted frame indicates the location of predicted co-inducer binding site at the interface of two monomers. **c**, The 3D structure of DmdR predicted by AlphaFold2 and potential binding sites of acryloyl-CoA in its E-O domain. **d**, Relative transcript level of *dmdA* in wild type *R. pomeroyi* str. DSS-3, $\Delta dmdR$ and $\Delta dmdR$ complemented with wild type or site-directed mutants of *dmdR* in minimal medium with 6 mM DMSP as carbon source. Different lowercase letters above bars indicate significant differences at $p < 0.05$. The error bar represents standard deviation of triplicate experiments.

10. Lines 238-242: Because high intracellular concentrations of DMSP are required, could this molecule also function as an osmoprotectant? Moreover, micromolar concentrations of extracellular DMSP have been reported to activate the demethylation pathway in Roseobacters (Wirth et al., 2020, Sperfeld & Narváez-Barragán, et al., 2024).

Response: Thank you for your comment. Indeed, DMSP is reported to function as an osmoprotectant and we believe that this, along with its catabolism, is likely the major reason DMSP is concentrated within marine bacterial cells. Principally for this study, we feel that DmdR-dependent gene regulation contributes to the intracellular concentration of DMSP, as detailed throughout our manuscript. Nevertheless, we have made this fact more prominent in our manuscript as follows:

-Lines 47-49: “Bacterioplankton, particularly **Roseobacters**, can import and concentrate dissolved DMSP to high millimolar levels¹³ allowing it to act as an antistress compound in e.g. **osmoprotection**^{10,14}.”

We also thank the reviewer for raising the two important references. Our data (Appendix Fig. 2c) confirm their findings that micromolar DMSP significantly upregulates *dmdA*, and further reveal that millimolar DMSP induces a far more dramatic response. To address this point directly, we have revised the text on lines 116-118 to incorporate the references of Wirth et al. 2020 and Sperfeld & Narváez-Barragán et al. 2024 to acknowledge the induction effect at micromolar concentrations.

Appendix Fig. 2 DmdR represses *dmdA-acuI* transcription in a DMSP and acrylate concentration-dependent manner. **a-b**, DMSP-dependent production of DMS (**a**) and MeSH (**b**) by wild type *R. pomeroyi* DSS-3, $\Delta dmdR$ mutant and genetically complemented ($\Delta dmdR/dmdR$) strains in minimal medium with 6 mM DMSP as carbon source. **c-d**, Relative transcript levels of *dmdA* in wild type DSS-3 cells grown with different concentrations of DMSP (**c**) and acrylate (**d**) compared to the CM treatment. **e-f**, Relative transcript levels of *acuI* in wild type DSS-3 cells grown with different concentrations of DMSP (**e**) and acrylate (**f**) compared to the CM treatment. **g**, Relative transcript levels of *acuI* in wild type (DSS-3), $\Delta dmdR$ and $\Delta dmdR/dmdR$ strains grown with different concentrations of DMSP and acrylate compared to the CM treatment. **h**, Upper panels: Coomassie brilliant blue stained SDS-PAGE gel to show the equal amounts of loaded protein for western blot analyses. Lower panels: Western blot analysis of DmdA in wild type DSS-3, $\Delta dmdA$ and $\Delta dmdA/dmdA$ cells grown with 6 mM DMSP, with coomassie brilliant blue stained gel to show the equal protein loading. **i-j**, Relative transcript levels of *dmdR* in wild type DSS-3 and $\Delta dmdR$ strains grown with different concentrations of DMSP (**i**) and acrylate (**j**) compared to the CM treatment. Error bars represent the standard deviation of triplicate biological replicates. A two-sided Student's *t*-test was used to assess statistically significant differences (**, $p < 0.001$; *, $p < 0.01$; *, $p < 0.05$; ns, $p > 0.05$). All experiments were carried out at least three times.

-Lines 116-118: “Transcription of *dmdA-acuI* was upregulated (< 20-fold) by DMSP or acrylate added to 300 μ M levels, consistent with the demethylation pathway being activated by micromolar DMSP concentrations in Roseobacters^{16,45}”

11. Lines 318-320: Are there examples where DMSP catabolites can be toxic? In addition, could the growth differences observed between $\Delta dmdE$ and $\Delta dmdF$ mutants and the WT on Succinate + DMSP reflect the ability of the WT to utilize both substrates as carbon sources, while the mutants cannot utilize DMSP (Fig. 6A)?

Response: Thank you for raising these important points. Regarding the potential toxicity of DMSP catabolites, we have cited relevant references in our manuscript. Acrylate and its downstream product acryloyl-CoA, generated via the cleavage pathway, are known to be toxic, as stated on lines 235-236 “Enhanced DMSP cleavage accelerates acrylate and subsequently acryloyl-CoA production, which are potentially toxic to cells^{38,44}”. Additionally, hydrogen peroxide (H₂O₂), produced during the demethylation pathway, is another toxic catabolite. This is described on lines 344-345 “Given H₂O₂ is generated by the demethylation pathway¹⁷⁻¹⁹, it makes sense for DmdF expression to be upregulated with DinB, previously shown to be involved in oxidative stress protection^{55,56}”.

We fully agree with your interpretation of the growth differences observed between the $\Delta dmdE$ and $\Delta dmdF$ mutants and the WT on succinate plus DMSP. It is likely that the WT, capable of utilizing both carbon sources, exhibits a growth advantage, whereas the mutants cannot utilize DMSP, leading to their compromised growth. We have now removed the overinterpreted conclusion that “...implying accumulation of toxic DMSP catabolites in these mutants” from the revised manuscript and sincerely thank you for highlighting this issue.

12. Lines 342-345: These are very interesting results, the discussion would benefit from including literature on DMSP demethylation and oxidative stress in bacteria. For example, in response to oxidative stress, *Ruegeria* DMSP metabolism has been reported to shift from the demethylation to the cleavage pathway, and KatG expression is influenced by DMSP exposure (Wang et al., 2022). In environmental datasets, *dmdA* and *katG* are reported to be co-expressed (Varaljay et al., 2015). Moreover, DMSP demethylases have been shown to be induced by oxidative stress in *P. inhibens*, which seems to be conserved in other bacteria (Narváez-Barragán et al., 2025). Since oxidative stress is also related to the switch between demethylation and cleavage pathways in *R. pomeroyi*, it would be helpful to test whether *dmdR* is upregulated by H₂O₂.

Response: Thank you for your constructive suggestion and for providing us these highly relevant references. In response, we have revised the manuscript to incorporate the key findings from suggested literature as follows:

-Line 350-355: “Since many DMSP demethylating bacteria lack DmdE and DmdF, we propose they, e.g. SAR11 and *R. pomeroyi* DSS-3, contain distinct peroxidases that are upregulated with DMSP demethylation. A potential candidate would be the catalase gene *katG* whose expression in *R. pomeroyi* DSS-3 is known to be induced by DMSP^{18,57}. Furthermore, *dmdA* and *katG* are reported to be co-expressed in environmental datasets⁵⁸”

-Line 68-69: “DMSP catabolism is often inducible by DMSP, its catabolites or by both of these^{10,23,24,38,40}, but it can also be regulated by oxidative stress⁴¹”

To determine whether *dmdR* is upregulated by H₂O₂, we exposed wild-type DSS-3 to 0.5 mM H₂O₂ and collected samples at different time points. RT-qPCR analysis revealed that H₂O₂ addition decreased *dmdR* transcription. To investigate whether this repression was specific to *dmdR* or part of a general stress response, we also measured the transcript levels of *dmdA* and *dmmA*, the latter encodes dimethylamine monooxygenase, which is unrelated to DMSP catabolism. Both genes exhibited transcriptional profiles similar to that of *dmdR*, indicating a broad downregulation under H₂O₂ stress. Furthermore, if the reduction in *dmdR* transcripts were specific, one would expect a consequent upregulation of its target gene, *dmdA*; however, this was not observed. Taken together, these results imply that H₂O₂ does not specifically regulate *dmdR* and that *dmdR* responds to DMSP rather than oxidative stress *per se*.

Figure for reviewers removed.

13. Line 352: Could the higher abundance of *dmdR/dmdF/dmdE* transcripts in the mesopelagic zone be related to oxidative stress or substrate availability at depth?

Response: We sincerely thank the reviewer for their question regarding the metatranscriptomic data. We agree that increased abundance of *dmdR/dmdEF* transcripts in the mesopelagic zone could plausibly be linked to factors such as oxidative stress or substrate availability. However, as this finding is based solely on bioinformatic analysis with no process data, we would prefer to refrain from including this speculative interpretation in the manuscript.

14. A brief discussion on the implications of this regulatory switch toward the cleavage pathway for sulfur fluxes would strengthen the broader context of the study.

Response: We thank the reviewer for their useful comment and have added a brief section on the implications of this study on DmdR as follows:

-Line 413-420: “This model for DmdR function implies that flux through DMSP cleavage should initially be greater than demethylation in marine Roseobacters. This is quite the opposite to what is reported in previous studies where demethylation dominates DMSP degradation *in situ*^{8,15,52}. Further work on model organisms and diverse environmental samples in carefully controlled settings, which are closer to physiological conditions, are essential to better understand DmdR’s function in the environment. Nevertheless, there is no doubt DMSP cleavage enhances DMSP demethylation activity.”

Dear Prof. Zhang,

Thank you for submitting your manuscript for consideration by the EMBO Journal. It has now been seen by the original referees whose comments are enclosed. As you will see, they express interest in your manuscript and are broadly in favour of publication.

Yet, our editorial assistance team flagged multiple issues that must be thoroughly addressed before the paper can be accepted.

Thank you for the opportunity to consider your work for publication. I look forward to your revision.

Yours sincerely,

Yehu Moran
Academic Editor
The EMBO Journal

Read our guidance for manuscript revisions and related editorial policies: <https://link.springer.com/journal/44318/submission-guidelines#cms-Revised-submissions>

<https://media.springernature.com/original/springer-cms/rest/v1/content/27825798/data/v1>

- a point-by-point response to the referees' comments, with a detailed description of the changes made (as a word file).
- a word file of the manuscript text.
- individual production quality figure files (one file per figure)
- a complete author checklist
- Expanded View files (replacing Supplementary Information)
- a Reagents and Tools Table as part of the Methods section

Please remember: Digital image enhancement is acceptable practice, as long as it accurately represents the original data and conforms to community standards. If a figure has been subjected to significant electronic manipulation, this must be noted in the figure legend or in the 'Methods' section. The editors reserve the right to request original versions of figures and the original images that were used to assemble the figure.

We realize that it is difficult to revise to a specific deadline. In the interest of protecting the conceptual advance provided by the work, we recommend a revision within 3 months (10th Mar 2026). Please discuss the revision progress ahead of this time with the editor if you require more time to complete the revisions.

comments by editorial assistance team

*DATA AVAILABILITY SECTION: If this data is not yet public, please make it public before submission.

If your study doesn't include such data please use the following instead:

"This study includes no data deposited in external repositories"

*FUNDING: W2441012, 32330001, 92251303, 42276102 and ZR2024YQ076 are listed in the Acknowledgements but have not been entered into our system, please correct.

*Author Contributions: Please remove the Author Contributions section from the manuscript text and make sure that the contributions are correctly entered into our submission system.

*DisclCIS: Please correct the heading to "Disclosure and Competing Interests Statement"

*REFERENCE FORMAT: In the text of the manuscript, a reference should be cited by author and year of publication; no more

than two authors may be cited per reference; 'et al' should be used if there are more than two authors (i.e. Smith & Jones, 2003; Smith et al, 2000). In the reference list, citations should be listed in alphabetical order and then chronologically, with the authors' surnames and initials inverted; where there are more than 10 authors on a paper, 10 will be listed, followed by 'et al.'

*APPENDIX 1 FILE WITH Table of Contents: Please add page numbers to the table of contents in the appendix and correct the nomenclature to "Appendix Table S1" etc. and "Appendix Figure S1" etc. throughout.

*REAGENT TABLE: Please remove the reagents and tools table from the manuscript text and upload it as a separate file. There is a citation for a Table EV6 - should this be Appendix Table S6? Please correct.

*SYNOPSIS IMAGE: not provided. Please provide according to instructions.

*SYNOPSIS TEXT: not provided. Please provide according to instructions.

*FIGURE CALLOUTS: there are callouts for Fig 6K and Fig 7D but these does not exist. Please fix.

IMPORTANT: Results and Discussion are combined here, but this is not a report and there are 6 figures. These should be clearly separated into two sections clearly marked as "Results" and "Discussion".

- Figure legends:

1. Please note that the legend for figure 6 is not provided in the sequential manner. This needs to be rectified.

2. Please note that the exact p values are not provided in the legends of figures 2c; 4b. Please provide.

3. Please note that the error bars are not defined in the legends of figures 6a-f. Please define in the legends.

4. Please note that the measure of center for the error bars needs to be defined in the legends of figures 1c-h; 2a-c; 4a, b. Please define in the legends.

Referee reports

Referee #1:

The authors addressed all of my comments.

I recommend this paper for publication.

Referee #2:

The present study by Fu & Wang et al. focuses on the abundant marine metabolite DMSP, which can be utilized by marine bacteria as a source of reduced carbon and sulfur through demethylation and cleavage pathways, respectively. Although these pathways are known to be influenced by DMSP availability, the molecular mechanism controlling this metabolic switch has remained unclear. The authors identify the regulator DmdR as the key factor responding to changing DMSP levels and coordinating the shift between demethylation and cleavage in the model Roseobacter *Ruegeria pomeroyi*. They also uncover additional components of this regulatory system, including genes potentially involved in oxidative stress detoxification, and demonstrate the widespread presence of these elements in marine bacteria, highlighting their broader ecological relevance. The findings are well supported by the data, which are presented through several complementary approaches, and the revisions have improved the clarity and accuracy of the manuscript.

The experiment conducted in response #12 provides strong support for the hypothesis that DmdR responds directly to DMSP rather than to oxidative stress itself. It also adds further clarity to the discussion of how DMSP catabolism can be induced by both DMSP and its catabolites, while oxidative stress may still influence the system. Including these new data and discussing them in the manuscript would reinforce the authors' conclusions and further strengthen the study.

Referee #2

The experiment conducted in response #12 provides strong support for the hypothesis that DmdR responds directly to DMSP rather than to oxidative stress itself. It also adds further clarity to the discussion of how DMSP catabolism can be induced by both DMSP and its catabolites, while oxidative stress may still influence the system. Including these new data and discussing them in the manuscript would reinforce the authors' conclusions and further strengthen the study.

Thank you for your constructive advice. We have included the result that DmdR did not respond to oxidative stress into the manuscript in a new results section entitled “The influence of oxidative stress on *dmdR* and *dmdA* transcription” on Lines 376-387. This section is pasted below for ease.

The influence of oxidative stress on *dmdR* and *dmdA* transcription

Given the association of DMSP demethylation (Eyice et al, 2018; Schäfer et al, 2019; Wang et al, 2022) and DmdR to oxidative stress and its amelioration, respectively, the impacts of H₂O₂ addition on *dmdR* and *dmdA* transcription were examined in *R. lacuscaerulensis* ITI-1157 and *R. pomeroyi* DSS-3. RT-qPCR analysis revealed that H₂O₂ addition decreased both *dmdR* and *dmdA* transcription to similar extents (Appendix Figure S17). Moreover, the same decreased transcript profile was seen for *R. pomeroyi* DSS-3 *dmmA*, which encodes a dimethylamine monooxygenase entirely unrelated to DMSP catabolism. These data imply that oxidative stress does not specifically regulate *dmdR* or *dmdA*, and that *dmdR* responds to DMSP rather than oxidative stress *per se*. This may be different in other Roseobacters and further work is required to evaluate the impacts of oxidative stress on DMSP catabolism.

Dear Prof. Zhang,

I am pleased to inform you that your manuscript has been accepted for publication in the EMBO Journal.

You may qualify for financial assistance for your publication charges - either via a Springer Nature fully open access agreement or an EMBO initiative. Check your eligibility: <https://link.springer.com/journal/44318/how-to-publish-with-us>

Yours sincerely,

Yehu Moran
Editor
The EMBO Journal

Please note that it is The EMBO Journal policy for the transcript of the editorial process (containing referee reports and your response letters) to be published as an online supplement to each paper. If you should prefer removal of any referee-only figures included in the point-by-point response(s), e.g. because they may still be used for future publication or because they have been reproduced from published work by others, please do let us know immediately via response email.

More information is available here: <https://link.springer.com/partners/embo-press/editorial-policies#Peer%20review>